
# Analysing river network dynamics and active length - discharge relationship using water presence sensors

Francesca Zanetti[1], Nicola Durighetto[1], Filippo Vingiani[1], and Gianluca Botter[1]

[1]Department of Civil, Environmental and Architectural Engineering, University of Padua, Padua, Italy

**Correspondence:** Francesca Zanetti (francesca.zanetti@dicea.unipd.it)

**Abstract.** Despite the importance of temporary streams for the provision of key ecosystem services, their experimental monitoring remains challenging because of the practical difficulties in performing accurate high-frequency surveys of the flowing portion of river networks. In this study, about 30 electrical resistance (ER) sensors were deployed in a high relief $2.6~km^2$ catchment of the Italian Alps to monitor the spatio-temporal dynamics of the active river network during the fall of 2019. The

set-up of the ER sensors was personalized to make them more flexible for the deployment in the field and more accurate under low flow conditions. Available ER data were analyzed, compared to field based estimates of the nodes' persistency and then used to generate a sequence of maps representing the active reaches of the stream network with a sub-daily temporal resolution. This allowed a proper estimate of the joint variations of active river network length ($L$) and catchment discharge ($Q$) during the entire study period. Our analysis revealed a high cross-correlation between the statistics of individual ER signals and the flow

persistencies of the cross sections where the sensors were placed. The observed spatial and temporal dynamics of the actively flowing channels also revealed the diversity of the hydrological behaviour of distinct zones of the study catchment, which was attributed to differences in the catchment geology and stream-bed composition. The more pronounced responsiveness of the total active length to small precipitation events as compared to the catchment discharge led to important hysteresis in the $L$ vs. $Q$ relationship, thereby impairing the performances of a power-law model frequently used in the literature to relate these two

quantities. Consequently, in our study site the adoption of a unique power-law $L - Q$ relationship to infer flowing length variability from observed discharges would underestimate the actual variations of $L$ by $40\%$. Our work emphasizes the potential of ER sensors for analyzing spatio-temporal dynamics of active channels in temporary streams, discussing the major limitations of this type of technology emerging from the specific application presented herein.

## 1   Introduction

Headwater streams - as well as rivers located in semiarid regions - are often characterized by the presence of reaches (or river segments) where water doesn't flow permanently throughout the year. While the terminology might vary among different authors, these non-permanent rivers are typically referred to as temporary streams. Temporary streams are frequently classified into a number of different categories (e.g. intermittent, ephemeral, episodic, seasonal) depending on the underlying temporal patterns of flow persistency (Williamson et al., 2015; Skoulikidis et al., 2017; Costigan et al., 2016). In recent years, many

studies have emphasized the ability of temporary streams to perform unique biogeochemical functions and provide a number of





important ecosystems services, among which the transport of material or organisms that support the biodiversity of downstream ecosystems (Datry et al., 2014; Leigh et al., 2016; Stubbington et al., 2017; Acuna and Tockner, 2010). The development of specific laws governing the use of water in non-permanent streams would represent an important step forward in water policy, since the number and extension of temporary streams is likely to increase in the future due to the combined action

of urbanization, ground and surface water withdrawal and climate change (Creed et al., 2017; Jaeger et al., 2019; Ward et al., 2020). To raise awareness of the importance of temporary streams in the scientific community and the society, it is fundamental to provide the community with new data about network expansion and contraction, possibly exploiting recent technological advancements in instrumentation and models (Acuna et al., 2014; Whol, 2017).

Different types of measurements have been conducted over the years to monitor network dynamics (Bhamjee and Lindsay,

2011). In most cases maps of the active network were obtained from field surveys carried out under diverse hydrologic conditions. While on-the-ground inspections are especially suited to characterize monthly or seasonal variations of the wet length of small catchments (Day, 1980; Morgan, 1972), this method was also applied for the description of the effect of event-based rainfall variability on the spatial and temporal patterns of flowing streams (Durighetto et al., 2020; Jaeger et al., 2019; Jensen et al., 2018; Ward et al., 2018). However, this method proved to be highly time-consuming also for relatively small catchments.

Recent technological advances in the field of environmental sensing provide a good opportunity to support the observational reconstruction of stream network dynamics. The most widespread automatic techniques applied for the study of temporary streams include high-resolution aerial photographs, LiDAR data (Spence and Mengistu, 2016; Roelens et al., 2018) and temperature sensors (Constantz et al., 2001; Blasch et al., 2004). More recently, electrical resistance (ER) sensors have been also proposed as a new alternative to the already existing methods commonly used to detect spatiotemporal variations of active

channels. ER sensors are as cost effective as temperature sensors, and they can be used with high temporal resolutions (up to 1 measurement every 5 minutes), thereby enabling a proper assessment of the impact of short-term climate variability on the active channel length. Two main techniques are reported in the literature for the deployment of ER sensors. A first technique consists in manufacturing a sensor made up of two distinct parts: i) the head containing the electrodes, which is located on the channel bed; and ii) the logger used to measure and record the response of the sensor head, which is typically located

nearby (Bhamjee et al., 2016; Peirce and Lindsay, 2015; Assendelft and vanMeerveld, 2019). The second technique, instead, consists in converting already existing temperature sensors (Blasch et al., 2004; Adams et al., 2006; Jaeger and Olden, 2012) or commercially available temperature/light data loggers into ER sensors (Chapin et al., 2014; Goulsbra et al., 2014; Jensen et al., 2019; Kaplan et al., 2019; Paillex et al., 2020). For each specific case study, the set up of the deployment in the field was typically chosen based on the properties of the river bed and the related substrate (e.g. rock surfaces, soil, alluvial sediments,

meadows).

Despite the spread of use of electrical resistance sensors to monitor water presence in dynamical stream networks, the major practical difficulties implied by the deployment of ER sensors under different setups have been seldom discussed in the literature, and a flexible setup that can be suited to the heterogeneous substrates usually found in high relief headwater catchments is yet to be found. Moreover, in most cases ER sensors were designed to provide information on the flow conditions

experienced by a specific point within the cross section of a stream, and they were not able to keep track of the hydrodynamic





conditions along the whole perimeter of the cross section where the sensors were placed, unless the stream bed was properly reshaped to convey the entire water flow towards the sensors (Assendelft and vanMeerveld, 2019). Additionally, while ER timeseries were often used to represent the spatial and temporal evolution of the active network in dynamical rivers, the statistical properties of individual ER timeseries have never been compared with independent empirical estimates of the local
persistency of the channel segments hosting these sensors.

In the search of a mathematical synthesis of the coevolution of network dynamics and the hydrological response of catchments, observed variations of the active channel length were frequently compared to the corresponding discharge values observed at the catchment outlet (Godsey and Kirchner, 2014; Jensen et al., 2017, 2019; Lapides et al., 2021). This led to the formulation of a power-law model connecting the active channel length $L$ and the catchment streamflow $Q$, which was often
used to get a simple mathematical description of the hydrologic dynamics involved both in rainfall-runoff mechanisms and river network dynamics. While the $L$ vs. $Q$ power-law relationship is empirical, its parameters have been shown to bear the signature of major geomorphological traits of the contributing catchment (Prancevic and Kirchner, 2019). As of now, observed discharge vs. active length power-law relationships were mostly derived by observational data characterized by a relatively low temporal resolution (i.e. from weekly to seasonal) (Prancevic and Kirchner, 2019). As coupled high-frequency discharge and
active stream length dynamics were seldom observed empirically (Jensen et al., 2019), the suitability of empirical power-law models to describe event-based changes in the flowing length of a river network at different temporal resolutions still needs to be further investigated.

On this basis, we have identified the following research questions for this study: (1) Can we identify a set up for ER sensors suitable to the heterogeneous land covers of high relief headwater catchments, and capable of detecting water flows in any
portion of the stream section? And what are the major practical problems in the deployment of this type of ER sensors? (2) Is it possible to link the statistical features of ER signals with observational data about the flow persistency in individual nodes of the river network? (3) Are the goodness of fit of a power-law model that links catchment discharge and wet length and the associated model parameters constrained by the temporal resolution of the available data? These questions are addressed by coupling observed data obtained from a network of ER sensors placed in a small catchment in the Italian Alps, a series of
statistical analysis of the collected field data and some modeling exercises. The remainder of this paper is organized as follows. Section 2 describes the study catchment, the design of the water presence sensors and the type of analysis performed exploiting the available empirical data. Section 3 presents the main results of the study: the description of the time series of the electrical signals, the statistical analyses performed on the data, the reconstruction of the observed active network dynamics and the analysis of the relationship between active length and discharge. In Section 4 we discuss the main findings of this work, with
specific reference to the research questions of this study. A set of conclusions closes then the paper.



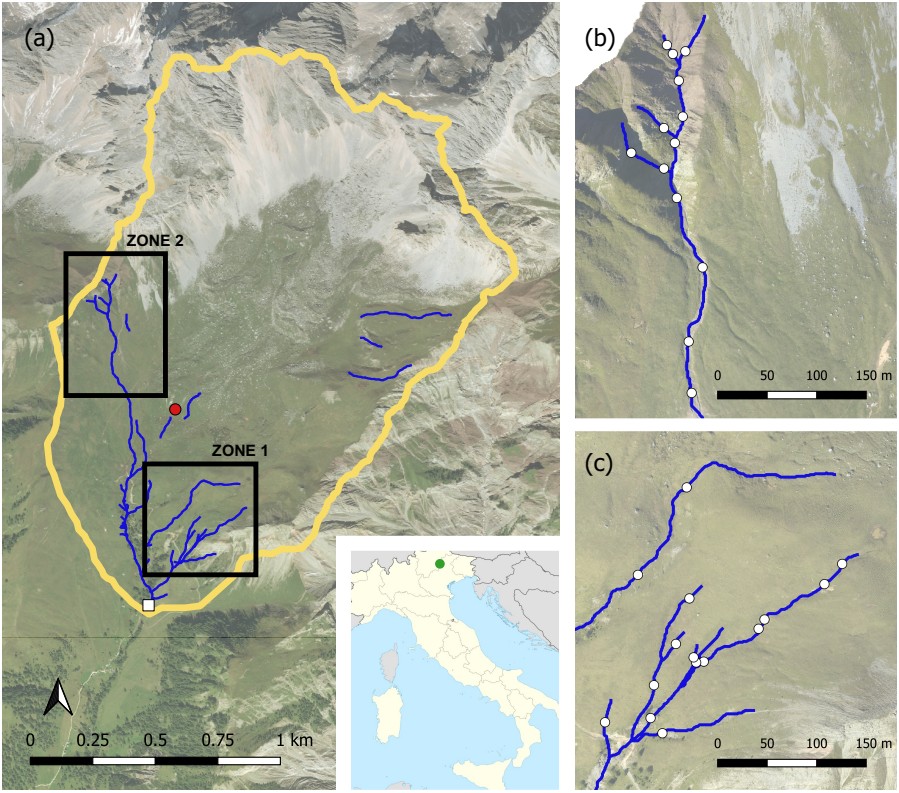

**Figure 1.** (a) Ortophoto of the Valfredda catchment and its location in northern Italy (green dot), position of the meteorological station (red dot) and of the section where discharge measurement were taken (white square). Sensors placed along the tributaries of Zone 1 (c) and Zone 2 (b).

## 2 Materials and Methods

### 2.1 Study site

This study took place in an alpine creek located in North-Eastern Italy, the Rio Valfredda (Figure 1). In particular, the test catchment comprises the northern part of the Rio Valfredda, which has a total contributing area of 2.6 km$^2$ and is charac-

terized by an average annual rainfall of approximately 1500 mm, with most of the precipitation concentrated between April and August, while during winter the catchment is usually covered by snow. Temperatures vary among seasons, in 2019 the minimum was recorded in January (-11.9°C) and the maximum in July (30°C). The catchment area spans a wide range of elevations (between 1900 and 3000 m a.s.l) and is characterized by heterogeneous morphological traits. On the upper part of the catchment, deposits of gravel and rocky debris are predominant. These deposits are covered by thin grasslands and ensure

a high soil permeability. Below 2100 m a.s.l. trees grow mostly along the streams on a sedimentary bedrock (Durighetto et al.,





2020). This heterogeneity of the landscape strongly influenced the observed hydrological dynamics and placed a constrain on the experimental set-up of the study.

## 2.2   Water presence sensors

The dynamics of the active stream network during the study period (September and October 2019) were observed using 31
onset HOBO Pendant loggers (HOBO UA-002-64, Onset Computer Corp, Bourne, MA, USA, hereafter HOBO) suitably modified following the methodology suggested by Chapin et al. (2014). The main changes introduced in this study consisted in removing the light sensor and adding two long electrodes, which recorded a positive electrical signal when connected by the flowing water. The obtained electrical resistance (ER) sensors were placed along the river networks to estimate flow intermittency within different network nodes. The electrical conductivity signal recorded by the HOBO ranged from 0 to
330000 lux (corresponding to the maximum value recorded by the sensors when the two electrodes were fully immersed in water) and the temporal sampling resolution was set to 5 minutes.

The river bed of the Valfredda creek is highly heterogeneous and hydro-morphodynamic variations induced by changes in flow magnitude might cause the flowing water to dodge the sensors, thereby impairing the reliability of the recorded data. To avoid cases in which water flow paths could bypass the sensors, particularly during low flow conditions, a novel experimental
set up was identified. It allowed the monitoring of the hydrological state of the whole cross-section where the sensors were placed. The two machine pin electrodes coming out of the sensors' housing cap were connected with two stainless steel wires rolled up to a geotextile net from 50 cm to 100 cm long. The length of the geotextile and wires guaranteed that the entire cross-section of the channel was connected to the sensor avoiding interference due to possible variations of the flow field. Rolling the cables on the network prevented them from moving when flooded, possibly creating artificial short-circuits or by-passes
that might impair the reliability of the electrical signal recorded. The net was then attached to another geotextile using plastic buttons to separate the electrodes from the ground by a few millimeters (and avoid interference with the wet soil). When placed in field, HOBOs were secured to their networks with suitable plastic strips; silicon was used to protect all the sensors' housing caps and prevent infiltration of water within the sensors.

The water presence sensors were installed mainly into two different regions of the test catchment, as shown in Figure 1. 15
sensors were deployed along tributaries of the southeastern part of the basin (Zone 1), an area that has glacial morphogenesis and is characterized by moraine deposits shaped on mild slopes covered by pastures (Figure 2a); therein, the river width ranges from 20 to 80 cm and the sensors were fixed to the ground using pickets (Figure 2c). The location of each sensor was chosen based on field surveys carried out prior to the installation of the instruments. ER sensors were placed in sections characterized by heterogeneous persistencies so as to avoid redundancy in the data. This led to a quite uneven spatial distribution of the
sensors, with a mean inter-sensors distance of approximately 55 m. 13 sensors were placed in the northwestern part of the network (Zone 2), where the riverbed is on a steep canyon composed of quartz porphyry rocks (Figure 2b). These rocks are small and unstable as they can move along the channel in response to rainfall events, thereby supporting the intermittent nature of hydrologic flows. In this region, the channel width ranges from 20 cm to about 1 m and the sensors' net were screwed on rock emergencies (Figure 2d). In zone 2, the criteria used in the selection of the HOBOs' positions were the same as for zone





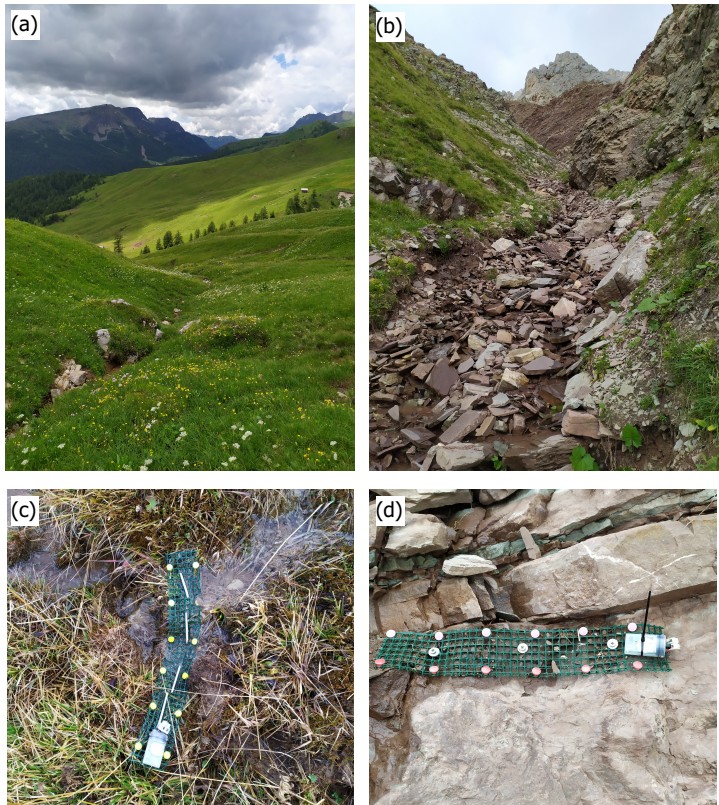

**Figure 2.** Left: picture of Zone 1 (a) and a sensor on the grass (c). Right: picture of Zone 2 (b) and a sensor screwed on a rock (d).

1. However, in this case a more even spatial distribution of the ER sensors was obtained, with a smaller mean inter-sensors distance (30 m). Other 3 sensors were installed along three disconnected branches of the network on the grassland between Zone 1 and 2. Data collected from the latter sensors were used only for analyzing the underlying network dynamics and not for the statistical analysis of Section 3.2 because they have experienced very few wet/dry transitions. Some of the tributaries of the Valfredda creek were monitored only through visual inspection, because field surveys allowed a simple yet reliable

characterization of the hydrological conditions experienced by those reaches during the study period (as they were permanently wet even under extreme low flow conditions or they were completely dry even during the most intense precipitation event of the entire period). The field surveys are described in the following sections.

## 2.3   Data analysis

### 2.3.1   Water presence data and flow persistency

In this study, ER data were collected between the 4[th] of September and the end of October 2019, before a major snowfall occurred that covered the catchment area and all the instruments. The data collected by the HOBOs were analyzed and two





hydrologically-relevant indexes were calculated from the available time series: the *average intensity (AI)* and the *exceedance of the threshold (E)*. *Average intensity* is the mean of the electrical signal registered by each HOBO in the period of record. *Exceedance of the threshold* is the probability that the electrical signal registered by a sensor is greater or equal than a chosen threshold value, ideally separating the wet condition from the dry condition. Different thresholds were initially considered but the final value (270000 lux) was chosen based on the intensity measured by the instruments placed in wet nodes during the field surveys. We decided to infer the threshold value directly from field observation instead of relying on laboratory experiments (e.g. with a soil column) to get a more reliable representation of the complexity of the natural environments typically found in most headwater catchments of the Alpine region. The above hydrological indexes ($AI$ and $E$) were calculated for each ER sensor and subsequently correlated with the persistency of the corresponding nodes, which was estimated as detailed below.

During the study period several field surveys were conducted to check the reliability of the signal recorded by the ER sensors, download the data and monitor the status of the network nodes under different hydrological conditions. However, these surveys were not homogeneous in space and systematic in time, thereby making impossible a reliable estimation of the persistency of the network nodes during the study period on a purely experimental basis. Therefore, we decided to estimate the persistency of the nodes using a combination of field observations carried out between 2018 and 2020 and some modeling. In particular, we used a model that links the spatial configuration of the network to rainfall data, as detailed in Durighetto and Botter (2021 (in press). The model was calibrated and validated based on 24 complete field surveys carried out for the whole Valfredda catchment from the Summer of 2018 to the Fall of 2020. During each survey, more than 500 nodes were classified as $wet$ or $dry$ based on the observed hydrologic conditions of the network (Appendix A). The main model assumptions and its performance in reproducing observed stream dynamics are detailed in the Appendix C. The simulation of network expansion and contraction was then used to calculate the persistency of the nodes in the study region. The latter was calculated as the fraction of time during which a node was simulated as active in the reference period (September and October of 2019). Previous studies have indicated that the model is able to accurately reproduce the observed spatial patterns of persistency in the study catchment under different hydrological conditions (Durighetto et al., 2020; Botter and Durighetto, 2020).

### 2.3.2 Rainfall and discharge data

Discharge measurements were taken in a cross-section at the outlet of the study catchment where water flows permanently. A pressure transducer allowed the measurement of the water stage with a temporal resolution of 5 minutes. Point stream flow measurements were combined to stage data to estimate the rating curve at the outlet of the catchment. Discharge data were collected with a three dimensional flow tracker under different hydrologic conditions so as to derive a reliable rating curve. Discharge time series were then derived for the entire study period with a temporal resolution of 3 hours, so as to reduce the noise in the recorded signal.

To visualize the hydroclimatic dynamics experienced by the study catchment during the focus period, we used rainfall data gathered from a meteorological station which was installed in 2018 within the basin in the context of the ERC-funded "DyNET" project (Figure 1). The meteorological station is used to monitor precipitation, temperature, relative humidity, net solar radiation and wind speed with a sub-hourly temporal resolution.





### 2.3.3 Corrections applied to the ER data

A common problem affecting the data recorded by the water presence sensors of this study is that the probes could not be all deployed in the field at the same time. Moreover, several sensors didn't work properly during their deployment. In particular, two HOBOs placed on the grass (Zone 1) exhibited a tendency to silt while others were found detached from their geotextile nets because of the presence of several horses grazing in the area. In Zone 2, debris flows, triggered by precipitation, prompted the accumulation of wet sediments around the geotextile altering the signal recorded by the electrodes of a probe placed along the rock canyon. Thus, for both reasons there were many missing data in the time series that had to be dealt with. In Appendix B we describe all the corrections that were applied to the data to take into account the lack of synchronicity of the available water presence data and the technical problems encountered during the deployment.

### 2.4 Spatial and temporal dynamics of the active network

A visual representation of spatial and temporal dynamics of the river network was obtained based on the intensity signal recorded by the HOBOs, as detailed below. For each time of the surveyed period, the electrical signals recorded by each sensor and its neighbours were interpolated in space in order to define which part of the stretch connecting the nodes was wet or dry. The threshold value of the electrical signal used to determine the wet or dry status of the stretches was 270000 lux, as detailed above (§ 2.3.1). Probes were divided into three categories depending on the electrical signal recorded:

- *Active*, when the value of intensity was greater than the threshold and the cross-section where the HOBO was placed was identified as wet.

- *Inactive*, when the signal collected was lower than the threshold and the cross-section was identified as dry.

- *Missing data*, when the sensor didn't provide reliable data (§ 2.3.3).

We identified a river stretch as a reach connecting two subsequent HOBOs. When two neighbouring sensors were both active, inactive or missing data, the stretch in between was defined as active, inactive or missing data accordingly. Instead, if two neighbouring HOBOs had data, but one of them was active and the other was inactive (or when a sensor with missing data had two neighbouring HOBOs providing reliable data), the wet length in the stretch connecting those sensors was calculated using a linear interpolation of the electrical signal measured by the nearest probes. Whenever a stretch had both missing data sensors at its end points and it was located between two concurrently active or concurrently inactive stretches, it was classified as active or inactive accordingly. Instead, if a stretch with missing data was located between two stretches with a different status (i.e. one active and one inactive), it was plotted as a stretch with missing data - as the data available did not allow a proper identification of the status of the sensors and the location of the wet-dry transition. This method couldn't be applied to stretches classified as missing data if they were located at the sources or in presence of confluences. In these cases the behaviour of the neighbouring stretches, experimental evidences, and the observed spatial patterns of persistency were considered in order to define the status of the stretch (active, inactive or missing data). Finally, a time-lapse visualization of the stream network dynamics with a temporal resolution of 3 hours was obtained using a MATLAB code.





## 2.5 Active length vs. discharge power-law model

Catchments can be seen as dynamical systems where the discharge at the outlet and the active length co-evolve in time in response to the underlying climatic forcing. In this study, we seek to use high frequency ER data to evaluate the robustness of an empirical model often proposed in the literature, linking the wet length and the corresponding discharge using a power-law relationship (Godsey and Kirchner, 2014; Jensen et al., 2017; Lapides et al., 2021):

$$L = aQ^b \tag{1}$$

Operationally, the analysis was performed as follows. Logarithms of synchronous observations of L and Q were plotted in a Cartesian plane and a linear regression was applied to the data. The equation of the interpolating line corresponded to the linearization of the power law model:

$$log(L) = log(a) + b\,log(Q) \tag{2}$$

The parameters $log(a)$ (vertical intercept) and $b$ (slope) were first calculated for the whole set of available $L$ and $Q$ data with a temporal resolution of 3 hours, and the goodness of fit of the linear regression to the data was evaluated through the coefficient of determination, $R^2$.

Afterwards, to understand how the parameters of equation (1) and the goodness of fit of the linear regression depend on the temporal resolution of the available data, we performed a statistical resampling of the data as detailed below. First, we subdivided the study period into non-overlapping sub-periods with a constant duration equal to $T$ days and we extracted a single random date and time within each sub-period. Then, the values of $Q$ and $L$ observed during the extracted set of dates and times (one pair for each period with length $T$) were selected from the available time-series of discharge and active length. The linear regression given by equation (2) was applied to the resampled $Q$ and $L$ data in order to calculate the corresponding values of $R^2$ and $b$. This resampling was assumed to be representative of a sequence of surveys performed on a random date and time, with a mean interarrival equal to $T$ days. The random extraction was repeated 50 times and the mean value and the standard deviation of $R^2$ and $b$ ($<R^2>$ and $<b>$, respectively) were calculated. All the above operations were repeated using different values of $T$: 1, 2, 4 and 7 days. The mean values of $b$ and $R^2$ (and the corresponding standard deviations) were analyzed and plotted as a function of the mean sampling interarrival, $T$.

## 3 Results

### 3.1 Time series of water presence and electrical resistance

In most cases, time series of the electrical signals recorded by the sensors placed in the field, $I(t)$, show a pronounced temporal variability within the whole period of record. The analysis of the time series of the electrical signals recorded by the HOBOs elucidates the different hydrological behaviour of the two study zones, but also emphasizes the heterogeneity of the signal recorded by different sensors within each zone.





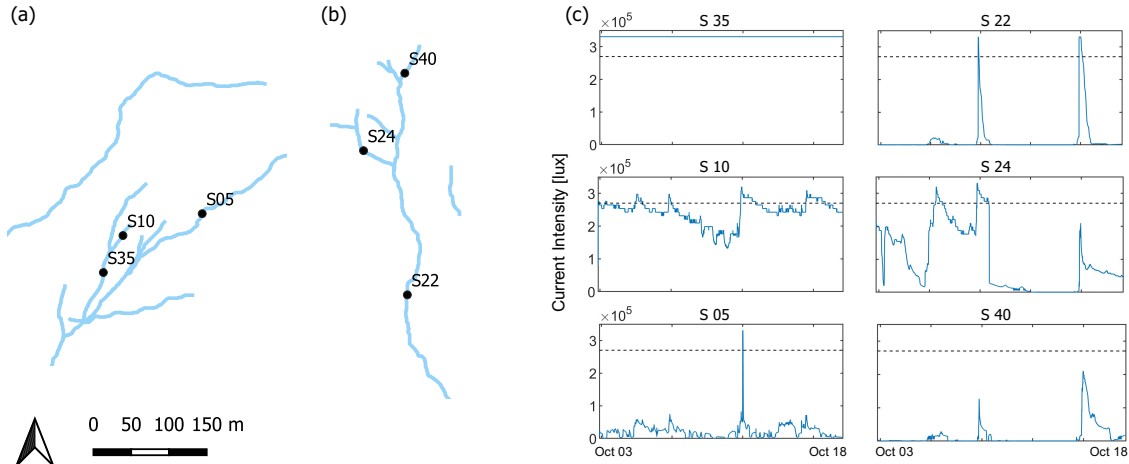

**Figure 3.** Examples of time series recorded by some of the sensors for two weeks between October 03 and October 18. The dashed line represent the threshold of 270000 lux (c) and location of these sensors along the tributaries of Zone 1 (a) and 2 (b).

The HOBOs belonging to Zone 1, which were all placed on a grassy substrate, exhibited quite heterogeneous behaviours. Some sensors, such as $S_{35}$ (Figure 2) systematically recorded intensity values close to the maximum intensity, because they were located along streams where water flowed permanently during the entire study period. Other sensors, instead, were placed in more dynamical streams and displayed an electrical signal that fluctuated between the maximum intensity (during rainfall events) and 100000-150000 lux (during the driest periods: $S_{10}$). Other sensors, mainly located in the higher part of Zone 1, recorded no intensity at all during most of the time, with intensity peaks over the threshold that were observed only during rainfall events, when discontinuous ephemeral ponds were generated in correspondence of the stream network ($S_{05}$).

Unlike the HOBOs of Zone 1, none of those placed in Zone 2 (Figure 2) was consistently wet during the whole study period. The probes activated during rainfall events and dried out afterwards with heterogeneous velocities depending on their position: sensors located in the central part of the creek persisted being wet for longer (see e.g. $S_{24}$) while those placed close to the channel heads turned off very quickly after each rain event ($S_{22}$). Other sensors, while they recorded an increase in the electrical signal during precipitation events, remained consistently below the 270000 lux threshold for the entire study period ($S_{40}$).

## 3.2 Statistical analysis

A statistical analysis was carried out to assess the consistency between the hydrologically-relevant indexes (namely, the *average intensity* and the *exceedance of the threshold*) calculated for each sensor, and the persistency of the corresponding nodes as derived from field surveys (§ 2.3.1). Figure 4 shows the scatter plots of persistency vs. *AI* and persistency vs. *E* for 28 sensors of Zone 1 and 2. All the available data were represented in the same two plots in order to emphasize differences and similarities among the HOBOs located in different zones.



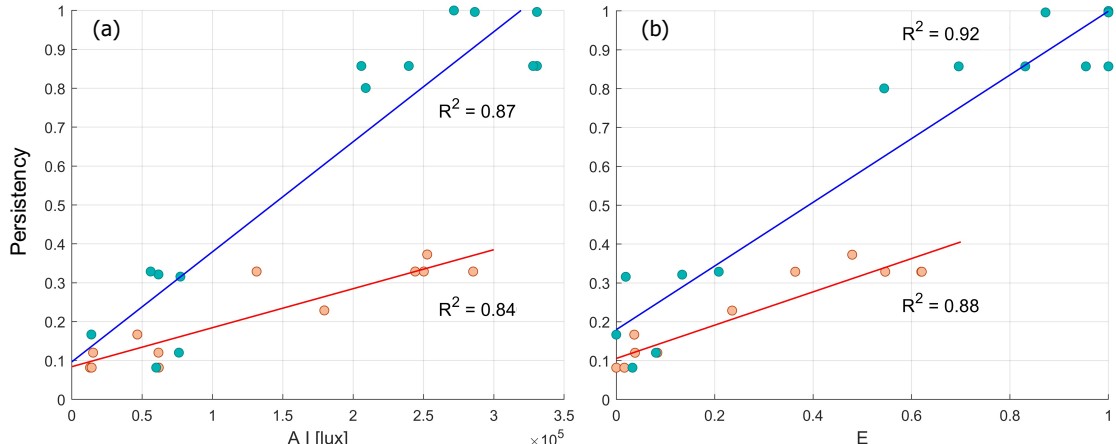

**Figure 4.** Scatter plots of persistency vs average intensity (a) and persistency vs exceedance of the threshold (b) of the sensors of Zone 1 (green points) and Zone 2 (orange points).

We hypothesize that an higher persistency of the nodes is reflected by an higher average intensity of the current recorded by the sensors and an enhanced probability that the electrical signal exceeds the selected saturation threshold. These hypotheses seem to be supported by the high coefficients of determination which were calculated from the data ($R^2$ always above 0.84).

However, Figure 4 also shows that the morphological characteristics of the studied zones (altitude and heterogeneous land cover) can influence the electrical signal registered by the instruments. Data collected from sensors of Zone 1 are not homogeneous despite they are located close-by in the field. The HOBOs of Zone 1 are clustered in two separate groups (Figure 4): one cluster includes the sensors located downstream that are mostly wet ($E > 0.7$ and $AI > 2\ 10^5$ lux) and the other includes only the HOBOs located upstream that turn on only during rainfall events ($E \leq 0.2$ and $AI < 0.8\ 10^5$ lux). Instead,

the tributary along which sensors of Zone 2 are located completely dries down, as indicated by the low values of the maximum persistencies observed in this area. The data-points in this cases are evenly distributed along the regression line, indicating a more homogeneous internal distribution of the node persistencies. It is also interesting to note that the two zones display very heterogeneous regression slopes between mean intensity and persistency. Instead, the slopes of the regression lines between exceedance of the threshold and persistency are comparatively more similar in the two zones. This suggests that $E$ might be

a more robust indicator of the underlying hydrological dynamics experienced by the nodes in the network, regardless of the specific position of the sensors in the catchment.

### 3.3 Stream network dynamics

Data collected by the sensors deployed along the watercourse provide information about the high-frequency spatial and temporal dynamics of the stream network. The full video of the observed river network dynamics is shown in the SI, while the main

text presents a sequence of snapshots taken from the video that represent the temporal evolution of the spatial configuration of





active channels in the study catchment (Figure 5). Figure 5a represents the channel network during the most intense rainfall event of the period studied, on the 8[th] of September, when all of the sensors of Zone 1 (left) were wet (blue dots). Not all the sensors of Zone 2 (right) had already been installed at that time. This circumstance explains the reason why some stretches are represented as no data (white dots) in that region. Also in Zone 2, on September 8, most of the network streams were wet, with

the exception of two river segments (shown in orange) that remained dry during the entire study period.

After the rainfall event that was observed on September 8, 2019 the network kept drying out for several weeks, and then wetted up again owing to a rain event that was observed on the 25[th] of September (shown in Figure 5b). This precipitation event was less intense than the previous one and it was characterized by a lower antecedent precipitation. Consequently, some headwater branches did not get wet in both zones.

The snapshot in Figure 5c shows the network on October 5, three days after an isolated precipitation event that took place in between a relatively dry period. On that date, all the HOBOs of Zone 2 were dry, while only the downstream sensors of Zone 1 (those closer to the permanent part of the network) became wet. Figure 5c emphasizes once again the dynamical nature of the channel network in the study catchment, and the different hydrological behaviour of the two focus regions identified in this study.

**3.4   Active length - discharge relationship**

The available hydrologic data and the data derived from the water presence sensors were also used to study the relationship between discharge and active length in the Valfredda catchment. Figure 6 shows the temporal evolution of rainfall ($h$), discharge and active length during the entire study period. During the first precipitation event, the most intense of the period, observed variations of $Q$ and $L$ were both significant. After September 11, instead, the intensity of the events was smaller and discharge

variations were barely noticeable. However, ER sensors indicate that small rainfall volumes were able to activate several channels for some days, leading to noticeable changes in the total wet length $L$.

Figure 7 shows the joint changes in active length and catchment discharge observed during the five periods shaded in green in Figure 6. During the most consistent precipitation event of the period (September 8), the peaks of active length and stream flow were reached at the same time (Figure 7a). Afterwards, the discharge showed a non-monotonic behaviour with a small

second peak, while $L$ decreased consistently from 2070m (September 9) to 444m (September 23).

During the precipitation events that took place between the 25[th] and the 30[th] of September (Figure 7b) and between the 02[nd] and the 03[rd] of October (Figure 7c), the maximum active length was reached a few hours after each rain pulse - though in the absence of significant $Q$ variations. Afterwards, the wet length first experienced a rapid decrease (again without significant changes in the discharge) and then it remained almost steady while the discharge kept decreasing. A rainfall event preceded the

event shown in Figure 7b and it was responsible to increase the catchment moisture conditions prior to the considered event. Therefore, after the end of the rainfall input, the active length remained stable for some time before it started declining. The trend observed during the event occurred between October 02 and 03 (Figure 7c) is similar to that shown in Figure 7b, but in this case the soil was drier before the event and the precipitation was less intense, thereby inducing a quicker decrease of

**Figure 5.** Map of the active stream network of Zone 1 (left) and Zone 2 (right) on the 8th of September (a), on the 25th of September (b) and on the 5th of October 2019 (c). Active stretches and sensors are blue, inactive are orange and no-data elements are white.

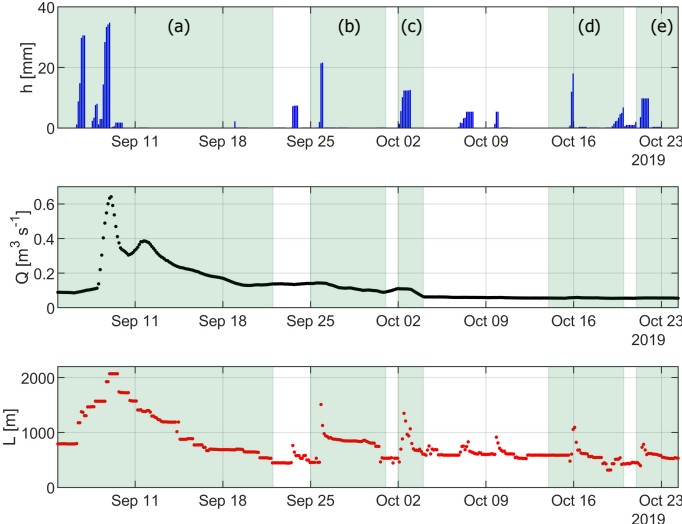

**Figure 6.** Top: temporal evolution of rainfall during the study period. Centre: temporal evolution of catchment discharge, $Q(t)$, during the study period. Bottom: temporal dynamics of active length, $L(t)$, during the study period. Areas in green represent the periods during which active length and discharge are plotted in Figure 7.

the active length as compared to what observed in the recession between 09/25 and 09/30. In both cases, however, consistent
variations of the wet length ($\Delta L \simeq 1 km$) corresponded to comparatively small variations of the discharge ($\Delta Q \simeq 50 l/s$).

The rain events that took place between the 14[th] and the 18[th] of October (Figure 7d) and between the 20[th] and the 24[th] of October (Figure 7e), instead, exhibited a different trend: the maximum wet length preceded the maximum discharge that was reached only when the active length was almost back to its initial value. In both these cases the intensity of the rainfall events was small and the variations of $Q$ were barely noticeable ($\Delta Q \leq 5 l/s$).

Figure 8 summarizes the joint changes of wet length and discharge during the whole study period. The variations of Q caused by the first precipitation event (blue dots) are larger than those observed in the remainder of the period. Instead, active length dynamics are observed during the entire study period. The available data of $Q$ and $L$ were fitted with a power law relationship, of the type shown in Eq. (1) (see Figure 8, where the joint changes of $L$ and $Q$ are shown in a log-log plot).

While we found that $log(L)$ and $log(Q)$ are linearly correlated ($p-$value below $10^{-5}$ at the 0.01 significance level), we also
observed a pronounced scatter of the points around the regression line, mainly due to the effect of small rain events. This scatter underpins the relatively low value of the coefficient of determination ($R^2 = 0.48$) of the interpolation obtained in this case.

It is interesting to note that the value of $b$ found in Figure 8 seems to be weakly impacted by the temporal resolution of the data used for the regression, as shown by the plot in Figure 9a. In particular, the maximum variation of $<b>$ obtained for different values of the mean resampling interarrival $T$ is around $1\%$. As expected, instead, the scattering of $b$ among different
realizations increases with $T$, owing to the different response of the $L$ and $Q$ signals to small precipitation events. The goodness





of fit shows a similar trend (Figure 9b), though the mean value of $R^2$ systematically increases with $T$ from $<R^2>= 0.485$ for $T = 1$ day to $<R^2>= 0.519$ and $<R^2>= 0.522$ for $T = 4$ and $T = 7$ days, respectively.

## 4 Discussion

Our personalized version of water presence ER sensors was successfully deployed in the field in two different regions of the
Valfredda catchment, which are characterized by heterogeneous substrates and different types of river bed. The sensors proved to be reliable in recording the hydrologic conditions of the whole cross section where the sensors were placed, especially during low flow conditions, when the observed stages were very low and the flow field was spatially heterogeneous and irregular within many cross sections. Nevertheless, our study indicates that water presence sensors require particular caution during their deployment in the field and the interpretation of the field data. The major issues that were faced during the deployment
changed based on the location where the HOBOs were placed: in the portion of the network dominated by debris (Zone 2) ER sensors were prone to be covered by sediments or flushed away by the flow field; conversely, in other regions of the network where the substrate was made by a thin organic soil covered by grass (Zone 1), the main problems were represented by grazing of animals and the formation of local pools with standing water during the drying of the network that could not be distinguished from flowing stream in the ER time series. In addition, in both regions there were issues related to the fragility of the steel
cables and/or the sealing system used to protect them from water infiltration during floods. Thus, intensive field surveys were necessary to check the functioning of the sensors and repair possible damages during the study period.

The high sampling frequency set for each instrument (5 minutes) and the large number of sensors used (more than 30 sensors for a maximum network length close to 2 km) allowed us to reconstruct maps of the active stream network with a high spatio-temporal resolution. The low cost of the HOBOs and their long-lasting batteries support their potential for monitor
active network dynamics in a wide range of contexts. However, while planning the use of these sensors it should be taken into account that some probes can be damaged or lost during the deployment and data acquisition, creating frequent gaps in the available time series.

Our data suggest that the two regions of the study catchments are strongly heterogeneous not only in terms of land cover but also in many features of the underlying network dynamics. The lower part of Zone 1 was permanently wet, with high values of
$E$ and persistencies close to 1 for most of the sensors. In the upper part of the same zone, instead, the network branches were more dynamical, and they usually got wet in response to precipitation events. Along the main channel of Zone 2, instead, water was able to infiltrate and exfiltrate rapidly creating frequent stream disconnections, promoted by the unstable morphology of the river bed. In this case, mean current intensities and node persistencies were consistently smaller than those observed in Zone 1.
In both regions, we found a good correlation between the persistency of the nodes and two statistical features of the electrical signal recorded by the ER sensors (namely, the mean intensity and the exceedance of a suitably selected saturation threshold). This suggests that the statistics of the ER signals recorded in the field could be robust indicators of the hydrological status of different network nodes. In particular, ER time series could be used to extrapolate information on the spatial patterns of node

**Figure 7.** Plots of discharge and active length during individual events across the study period, as indicated in Figure 5.





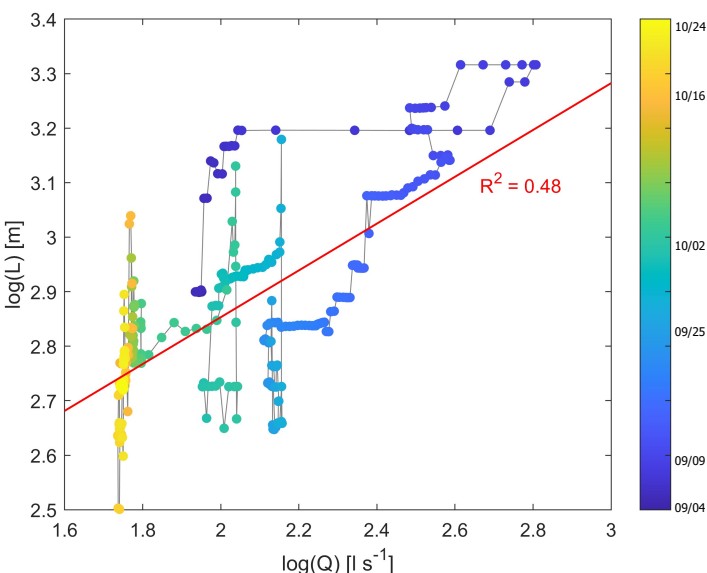

**Figure 8.** Plot of discharge and active length for the entire period studied.

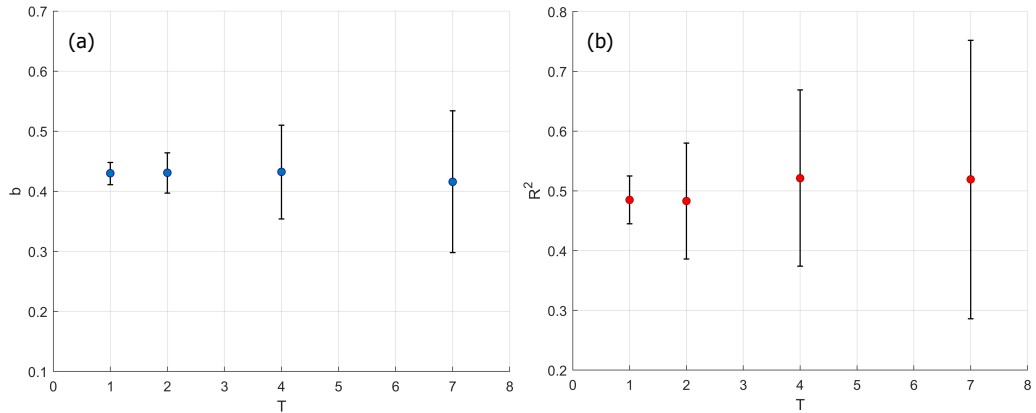

**Figure 9.** Plot of resampled $b$ (a) and $R^2$ (b) mean values and standard deviations as a function of the mean frequency of the resampling $T$.





persistency, which could be in turn used to define the hierarchy according to which the nodes in the network are activated in
response to rain events and then deactivated during the subsequent drying (Botter and Durighetto, 2020).

In line with the results presented by Jensen et al. (2019), our data indicate the presence of important hysteresis in the active
length vs discharge relationship at fine temporal resolution, especially during low flow conditions (Figure 8). Network length
was found to be more sensitive than discharge to small precipitation inputs: while most rain events induced visible changes
in the active channel length, the catchment stream flow was sensitive only to the rain events lasting for several consecutive
days (6-9/09, 13-18/10, 20-24/10) and to intense storms (more than 20-30 mm in 9-12 hours). We recognize the value of fitting
discharge and active length data with simple power law functions to analyse how stream network dynamics are constrained by
flow regimes in river basins. However, the low value of the regression coefficient emerging from our analysis ($R^2 < 0.5$, see
Figure 8) indicates that in some cases the use of a bijective $L - Q$ function to infer active length changes in catchments where
discharge time series are available, might lead to significant underestimation of the actual variations of the flowing channel
network. In our case study, the standard deviation of the wet length as derived from the sensors' data is 360 m, while the
standard deviation of $L$ predicted by the power-law model based on the observed variability of the discharges is only 224 m
(about $40\%$ lower). This underestimation is induced by the poor ability of the power law model to capture the observed network
dynamics produced by small precipitation inputs.

Interestingly, the resampling procedure (Figure 9a) indicates that the mean slope of the regression exponent, $b$, is not sig-
nificantly impacted by the specific sampling strategy adopted, thereby suggesting that the estimation of the parameter $b$ of the
power-law model could be performed exploiting data collected with a relatively coarse temporal resolution (e.g through weekly
surveys). Our analysis also suggests that when the frequency of the data is low, the hysteresis in the $L$ vs. $Q$ relationship are
on average less pronounced, leading to an increase of the mean goodness of fit for larger sampling inter-arrivals. Nevertheless,
the chances that the experimental $(Q, L)$ pairs don't exhibit a clear power-law trend also increases as the mean interarrival
between the observations used in the regression increases, as shown by the increase of the standard deviations of $R^2$ with $T$.
This suggests that the goodness of fit of the power-law model can be strongly dependent on the specific timing of the field
surveys in which active length and discharge are evaluated.

## 5    Conclusion

In this work, we have tested the use of ER sensors for the monitoring of active network dynamics in a high-relief 2.6 km$^2$
catchment of the Italian Alps during the fall of 2019. To this aim, we have utilized a personalized version of the HOBO sensors
previously proposed in the literature, which was modified to be suited for a deployment under different substrates and is deemed
to be more accurate under unstable hydrodynamic conditions and during low flow conditions. The following conclusions are
worth emphasizing:

– ER sensors provided precious information about high frequency space-time network dynamics in the study catchment
during the fall of 2019; in particular, collected ER data were used to produce a video and a sequence of maps representing
the dynamics of the active network with a sub-daily temporal resolution;





– The mean intensity of the ER signal, and the exceedance of a suitably selected intensity threshold were found to be highly correlated with the persistency of the network nodes. This suggests that ER sensors signal provides statistically meaningful information on the hydrologic status of different nodes of the river network;

– The successful application of ER sensors under the heterogeneous environmental conditions found in the Valfredda catchment suggests the good potential and the flexibility of the tool for river network mapping. Likewise, the study highlighted the major shortcomings of this type of technology and the dependence of these shortcomings on the type of substrate: in steep rocky channels the main problem is the sensor flushing during floods and the accumulation of sediments; in grassy regions the major problems relate to water ponding and grazing of animals that might remove the

sensors during the deployment. For the above reasons, we propose that the use of ER sensors for river network mapping needs to be quite intensively supervised;

– The high-frequency monitoring of flow rates and active stream lengths performed in this study allowed an in-depth analysis of the relationship between catchment discharge ($Q$) and active length ($L$) over a broad range of timescales. Our data highlighted the presence of important hysteresis in the high-frequency $L - Q$ relationship, mainly due to the

different responsiveness of the catchment streamflow and the active length to small precipitation inputs.

– The mean value of the exponent of the power law relationship between catchment discharge and total active length was found to be almost independent on the frequency of the observational data, which instead had a larger impact on the goodness of fit of the power-law model. When the frequency of the data is lower, the observed values of $R^2$ are, on average, larger but they are highly dependent on the specific times during which $L$ and $Q$ observations are taken.

*Data availability.* Data can be found on the online repository https://doi.org/10.25430/researchdata.cab.unipd.it.00000437

*Video supplement.* Two videos of the spatio-temporal dynamics of the river network of Zone 1 and Zone 2 can be found on the online repository https://doi.org/10.25430/researchdata.cab.unipd.it.00000437

**Appendix A: Flow persistency**

A conventional criterion was identified in order to classify as *wet* or *dry* the nodes along the intermittent tributaries of the

Valfredda creek during surveys in the field: if the stream flow had a width equal to or greater than 10 cm, the node was considered as wet (otherwise the node was identified as dry). This lay at the basis of the calculation of the active network length and the persistency of the nodes, as detailed in Section C.



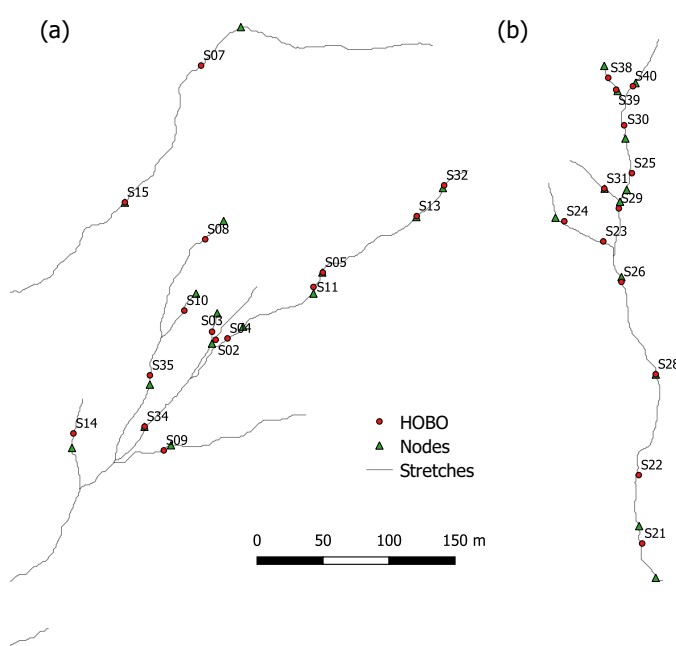

**Figure A1.** Nodes and corresponding sensors of the tributaries of Zone 1 (a) and Zone 2 (b).

The analysis of the nodes along the tributaries helped in the selection of the positions of the HOBOs to get a reliable representation of the underlying stream network dynamics (Figure A1). In doing that, a one-to-one association between the ER
sensors and some network nodes was also established.

**Appendix B:  Corrections applied to the data**

To account for missing data, weights ($\eta$) that relate the probability of exceedance of the threshold of all the active sensors of a zone to the exceedance of the threshold of a sensor during a period of time containing missing data, were calculated (Tables B1, B2).
The ratio between the probability of exceeding the threshold of a sensor during the period $t_A$, when it is not active, and $t_B$, when it is functioning, is assumed to be equal to the ratio between the mean probability of exceeding the threshold of all the other active sensors of the zone during $t_A$ ($P^{t_A}$) and $t_B$ ($P^{t_B}$) ($\eta$).

$$\eta = \frac{P^{t_A}}{P^{t_B}} \tag{B1}$$





| Sensor corrected | Missing data | Sensors used for the corrections | Data used | $\eta$ |
|---|---|---|---|---|
| $S_{29}$ | Sep 6th - Sep 11th | $S_{25}$ | Sep 11th - Sep 16th | 1 |
| $S_{25}$ | Sep 16th - Oct 18th | $S_{29}$ | Sep 11th - Sep 16th | 0.433 |
| $S_{21}$ $S_{38}$ $S_{39}$ $S_{40}$ | Sep 4th - Sep 11th | $S_{24}$ $S_{26}$ $S_{30}$ $S_{31}$ | Sep 11th - Oct 18th | 2.238 |
| $S_{22}$ $S_{23}$ $S_{28}$ | Sep 4th - Sep 16th | $S_{24}$ $S_{26}$ $S_{30}$ $S_{31}$ | Sep 16th - Oct 18th | 1.871 |

**Table B1.** Corrections applied to some of the HOBOs of Zone 2.

During rainfall events some of the sensors were buried by sediments but they kept measuring 330000 lux as if they were
completely wet even when there wasn't flowing water on the channel anymore. Their time series were reconstructed based on
those of the closest sensors not affected by sediment dynamics.

In some parts of the tributary of Zone 2 the channel bedrocks were particularly fragile and unstable and this prompted the
movement of debris along the channel and their deposition between a sensor and its geotextile network. In the upper part of
Zone 2, sensor $S_{39}$ was subject to frequent sediments depositions. During one of the surveys, it was found covered by debris
(Figure B1) and the luminous intensity that it registered was 330000 lux as if it was completely wet even though there was no
flowing water at the time. After cleaning the sensor (on the 7th of October), the measured value of electrical current dropped
drastically and, as shown by the time series (Figure B2), the intensity recorded by the sensor remained low until the following
rainfall event (October 9th).

The temporal dynamics of the electrical signal in the periods from the 26th of September to the 7th of October and from the
17th to the 18th of October, were reconstructed based on the time series of sensor $S_{38}$, not affected by sediment dynamics and
close to $S_{39}$. The recession rate with which the signal of $S_{38}$ declines after a rainfall event was estimated and that same rate was
applied to $S_{39}$ (Figure B2).

To correct the asynchronicity between the electrical signals recorded by the sensors and the persistencies of the corresponding
nodes, the model developed by (Durighetto et al., 2020; Botter and Durighetto, 2020; Durighetto and Botter, 2021 (in press)
was applied. It links the spatial configuration of the network to weather data and here it was applied to estimate the persistency
of the nodes between the 4th of September and the 24th of October 2019 knowing rainfall heights of that same period as well
as persistencies and precipitation data collected between July 2018 and January 2019 used to develop the model (Figure C1).

## Appendix C: Empirical model for local persistency

The local persistency of each node during the study period (4th September to 31th October, 2019) was estimated by means of
the field surveys carried out in 2018 and 2020 and the models described in Durighetto et al. (2020); Botter and Durighetto





| Sensor corrected | Missing data | Sensors used for the corrections | Data used | $\eta$ |
|---|---|---|---|---|
| $S_{04}$ $S_{13}$ | Sep $4^{th}$ - Oct $3^{rd}$ | $S_{02}$ $S_{07}$ $S_{09}$ $S_{11}$ | Oct $3^{rd}$ - Oct $24^{th}$ | 0.922 |
| $S_{05}$ | Sep $15^{th}$ - Oct $3^{rd}$ | $S_{02}$ $S_{07}$ $S_{09}$ $S_{11}$ $S_{14}$ | Sep $4^{th}$ - Sep $15^{th}$ | 0.667 |
| $S_{08}$ | Sep $15^{th}$ - Oct $3^{rd}$ | $S_{02}$ $S_{07}$ $S_{09}$ $S_{11}$ $S_{14}$ | Sep $4^{th}$ - Sep $15^{th}$ | 0.666 |
| $S_{10}$ | Sep $16^{th}$ - Oct $3^{rd}$ | $S_{02}$ $S_{07}$ $S_{09}$ $S_{11}$ $S_{14}$ | Sep $4^{th}$ - Sep $16^{th}$ | 0.663 |
| $S_{14}$ | Oct $17^{th}$ - Oct $24^{th}$ | $S_{02}$ $S_{07}$ $S_{09}$ $S_{11}$ $S_{35}$ | Sep $11^{th}$ - Oct $17^{th}$ | 1.097 |
| $S_{15}$ | Oct $8^{th}$ - Oct $16^{th}$ | $S_{02}$ $S_{07}$ $S_{09}$ $S_{11}$ $S_{33}$ $S_{35}$ | Sep $11^{th}$ - Oct $8^{th}$ | 1.129 |
| $S_{32}$ $S_{33}$ $S_{34}$ $S_{35}$ | Sep $4^{th}$ - Sep $11^{th}$ | $S_{02}$ $S_{07}$ $S_{09}$ $S_{11}$ $S_{14}$ $S_{15}$ | Sep $11^{th}$ - Sep $26^{th}$ | 1.300 |
| $S_{32}$ | Oct $16^{th}$ - Oct $24^{th}$ | $S_{02}$ $S_{07}$ $S_{09}$ $S_{11}$ $S_{35}$ | Sep $11^{th}$ - Oct $16^{th}$ | 1.099 |
| $S_{34}$ | Sep $26^{th}$ - Oct $18^{th}$ | $S_{02}$ $S_{07}$ $S_{09}$ $S_{11}$ $S_{35}$ | Sep $11^{th}$ - Sep $26^{th}$ | 0.981 |

**Table B2.** Corrections applied to some of the HOBOs of Zone 1.

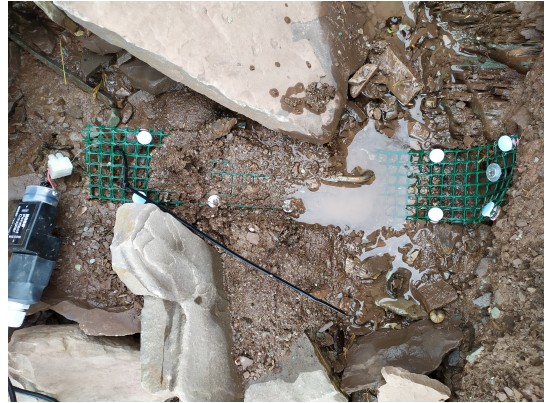

**Figure B1.** Sensor $S_{39}$ covered by sediments.



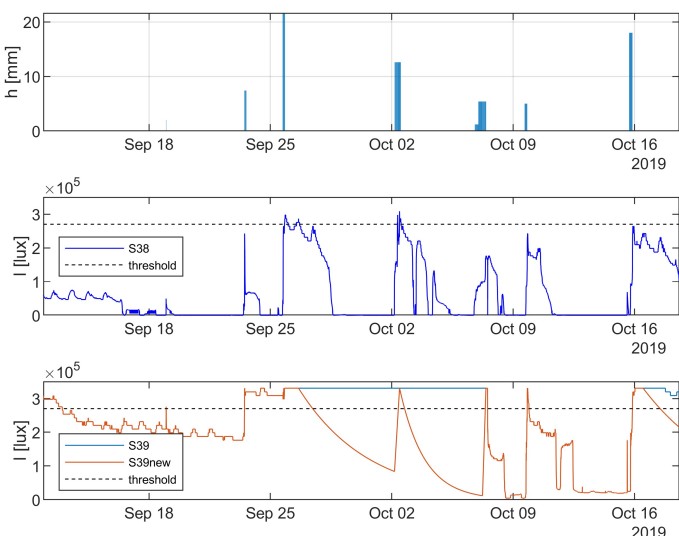

**Figure B2.** Top: hyetograph of the period studied. Center: time series of $S_{38}$. Bottom: time series of $S_{39}$ before the corrections (light blue) and after the corrections (orange).

(2020); Durighetto and Botter (2021 (in press). The procedure is composed of three steps. In the first step, the active length is estimated from antecedent precipitation via the equation:

$$L(t) = k_0 + k_5 h_5(t) + k_{35} h_{35}(t) \tag{C1}$$

where $L(t)$ is the active length, $h_5(t)$ and $h_{35}(t)$ are the antecedent daily precipitation accumulated over 5 and 35 days,
respectively, and $k_0, k_5, k_{35}$ are calibrated regression coefficients. The daily precipitation was measured by a weather station of the Veneto Region Environmental Protection Agency (ARPAV) located 4.5 km far from the catchment centroid. The coefficients were calibrated on the 9 field surveys carried out during 2018 (Mean Absolute Error = 1.1%) and validated with the 13 field surveys of 2020 (MAE = 3%). The simulated active length and the corresponding field surveys are shown in Figure C1 (for more information the reader is referred to Durighetto et al. (2020)).
The second step consists in simulating the spatial patterns of the active network, starting from the active length. This was accomplished exploiting the hierarchical model of network activation introduced by Botter and Durighetto (2020). The hierarchical model states that during network expansion nodes are activated in a given order (from the most persistent to the least persistent nodes in the network), and deactivated in the reverse order during network contraction. Botter and Durighetto (2020) proved the robustness of this model in the Valfredda catchment (average $F_1$ score equal to 0.98). The specific order of node
activation (and thus the hierarchy of the nodes) was identified from the field surveys. Then, for each day of the study period, the active nodes were determined by combining the hierarchical model and the empirical model for the active length.

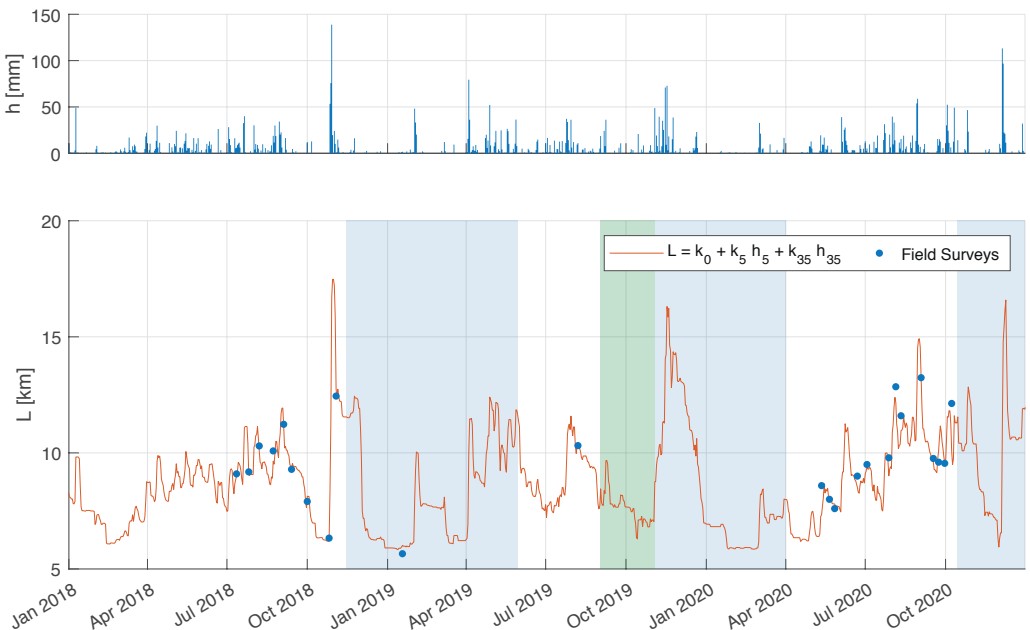

**Figure C1.** Timeseries of daily precipitation $h$, and the corresponding active length $L$ estimated by the empirical model in Durighetto et al. (2020). Blue dots show the 28 field surveys carried out on the catchment. The blue shade identifies periods with snow cover, while the green shade highlights the period during which the ER sensors were employed.

In the third and last step, the local persistency of each node was calculated as the fraction of time for which that node was active during the study period.

*Author contributions.* All authors carried out the field surveys during the study period. ND provided tools to study the data, FZ analyzed the results, wrote the video code and a first draft of the paper. GB identified the methods for the analysis and provided insight into data interpretation. All the author contributed to finalizing and editing the paper.

*Competing interests.* The author declare that they have no conflict of interest

*Acknowledgements.* We are grateful to the municipality of Falcade and the Compagnia della montagna di Valfredda - Mònt de le fède for making available the Valfredda catchment for this research project. We also thank Alfonso Senatore for the insightful comments. This study was supported by the European Research Council (ERC) DyNET project funded through the European Community's Horizon 2020 - Excellent Science - Programme (grant agreement H2020-EU.1.1.-770999).





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
