# Peer review of "Technical note: Analysing river network dynamics and active length – discharge relationship using water presence sensors"

_Hydrology and Earth System Sciences, 2021_

## Author Comment (AC2)

(a) September - October 2019

(b) July - November from 2010 to 2020

Figure 1: Frequency analysis of daily rainfall depth for the study period in 2019 (a) and for the years between 2010 and 2020 (b).

[Figure]

Figure 2: Relationship between the dynamical discharge (Qd) and network length(Ld) following the power law model with an initial b equal to 0.174 and different values of permanent discharge Qp and length Lp.

[Figure]

Figure 3: Relationship between the dynamical discharge (Qd) and network length(Ld) following the power law model with an initial b equal to 0.429 and different values of permanent discharge Qp and length Lp.

[Figure]

Figure 4: Relationship between the dynamical discharge (Qd) and network length(Ld) following the power law model with an initial b equal to 0.597 and different values of permanent discharge Qp and length Lp.

---

## Author Response (AR1)

**EDITOR**

*During the first discussion phase, the referees have provided a series of important comments and suggestions on your manuscript.*

We thank the editor and the referees for providing valuable comments which helped to reshape the paper. We have completely rewritten a significant fraction of the Ms. based on the feedback provided by the referees and the Editor - and we hope that the paper is now suited to be accepted for publication in HESS.

*Some of these relate to the specificity of the study area (questioning the appropriateness of the site for studying river network dynamics) [...]*

In the revised version of the Ms. we have discussed extensively the impact of the specificity of the site on the results presented in the paper. In particular we have provided a detailed analysis of the impact of the karst areas observed in the northern portion of the catchment on the $Q$ vs. $L$ relationship. Figures 8 and 9 have been modified accordingly. While extending our analysis to the dynamical components of $Q$ and $L$, we recognized a mistake in the original calculations. All the main results of the paper, however, remained unchanged, the only difference being the value of $b$, which turned out to be smaller than before. Moreover, in the revised text, we have also smoothed down our conclusions and we have explicitly acknowledged that our findings might not be generalizable to other sites with different characteristics.

*[...] the limited number of events [...]*

We have provided arguments to discuss the reasons that underlie the limited duration of the field campaign in this paper. Furthermore, we have also shown in the revised version of the paper that the study period is quite representative in terms of the underlying climatic conditions of the long-term behaviour of the Valfredda Creek. On the other hand, we have also mentioned the limitations of the study as implied by the limited duration of the study period.

*[...] related uncertainties and difficulties in drawing conclusions from the presented dataset (especially related to the relationship between stream length and discharge).*

In the revised version of the paper, we have discussed more carefully our results. We highlighted the fact that hysteresis in the $L$ vs. $Q$ relation have been already documented in the literature, we have explained the reasons underlying the observed hysteresis and how these reasons are not linked to the presence of the karst area in the northern part of the catchment. Finally we have emphasized that this site is highly heterogeneous, and the implied non-linearity in the hydrological response could enhance the scattering in the $L$ vs. $Q$ relationship.

*Consequently, I invite you to leverage your - for some aspects already very detailed - responses that you have provided to the referees' comments. Please consider all comments very carefully when preparing your revised manuscript.*

All the comments of the referees were very helpful in revising the paper. We would like to express our gratitude to the referees and the Editor for their work.

*Without encouraging you for one option or the other, I also invite you to carefully ponder the pros and cons of a re-orientation of the manuscript to what would be more of a technical note.*

Thanks for the suggestion. We thoroughly considered both possibilities, and we think that our analysis goes beyond a technical note, as several issues relevant for the observation and the modeling of intermittent streams are discussed in the paper. These issues are not limited to the use of ER sensors and the related pros and cons of this technique, but also include: i) the correlation between key statistics of ER sensors and the persistency of different nodes; ii) the reconstruction of high frequency space-time dynamics of the active stream network from ER data; iii) the analysis of the coupled dynamics of $Q$ and $L$ across a sequence of events and iv) the dependence of the parameters of the power-law model for $L$ and $Q$ on the sampling frequency. In the revised version of the paper we have tried to further stress the physical interpretation of our dataset. While we acknowledge that our results are far from being conclusive, we also think that all of this is interesting for the readership of HESS, and it requires more room that the few pages implied by a technical note.

**REVIEWER N.1**

**General comment**
*This is an interesting work based on a relatively new technology that has the potential to offer new insight into the dynamics of stream network in small catchments. Streams are highly dynamic systems and characterizing such "liveliness" is important to move towards a better understanding of catchments' functioning. Overall, this manuscript is well written, logically structured, and clearly illustrated. The introduction is solid and the results are well supported by the data. I have only some specific and minor comments that I would like the Authors to address (see below). Overall, I recommend a moderate revision but since the MODERATE grade is not available to HESS reviewers, I indicated minor review in the review form.*

We thank the referee for the positive assessment of our manuscript.

**Specific comments**
*L78-83. I think that the three research questions could be more appealing than the current ones. I understand that, as this is a relatively new measurement approach, aiming at providing the reader with some methodological observation is useful for possible further application of the methods. So, I like question 1. However, I would move it as last question because, in my opinion, it's more important to focus on processes that the method is able to describe and understand, rather than on the method itself. The second question is a bit too narrow because it implies a "yes" or "nor" and does not lead to much insight into hydrological processes. Similarly, question 3 sounds a bit too "methodological" and not so oriented towards process understanding. So, I suggest moving question 1 as third, and to rephrase question 2 and 3 and bit.*

Thank you for the suggestion. We rephrased the second and third research questions, and we changed their order as proposed. In particular, we reformulated the research question n.2 in order to avoid it to be a closed-answered question. This, in the revised text, has been rephrased as "how is the persistency of individual nodes of the network reflected by the statistical features of ER signals across a range of different substrates and various degree of flow intermittency?". Likewise the third question has been rephrased as follows: "how do wet length and catchment discharge co-evolve in response to a sequence of rain events, and how does their mutual relation depend on the temporal resolution of the available observations?". Moreover, in the discussion we went more into detail about the hydrological processes occurring in our study site (see lines 421-427) and how they affect the data collected in the field through our instruments and using the methods illustrated in the Ms.

*Indeed, my overall impression is that this work is much technically- and methodologically-oriented and less prone to describe and understand hydrological processes, and I guess that reader of this journal are more interested in knowing how catchment works rather than know if some data can fit a certain model or no. So, I invite the Authors to consider revising the work to reflect this aspect.*

Thank you for the comment. We trust our Ms is in line with the scope of HESS, as the aim and scope of the journal explicitly mentions "the study of the spatial and temporal characteristics of the global water resources (solid, liquid and vapour) and related budgets, in all compartments of the Earth system (atmosphere, oceans, estuaries, river, lakes and land masses), including water stocks, residence times, interfacial fluxes, and the pathways between various compartments". Moreover, in the aim and scope section of the journal it is explicitly claimed that "papers should contribute to the advancement of hydrological modeling, hydrological monitoring and data analysis, [...], experimental design and technology". For these reasons we believe that our work could be of interest for the readership of HESS. Furthermore, we would like to emphasize that we are aware of several highly cited, technically-oriented/methodological papers recently published by the journal (e.g. Kaplan et al., 2020 (DOI: 10.5194/hess-24-5453-2020); Kirchner 2016 part 1 (DOI: 10.5194/hess-20-279-2016) and part 2 (DOI: 10.5194/hess-20-299-2016)), an instance which makes us believe that being methodological does not represent per se a shortcoming for an HESS Manuscript. At the same time, we acknowledge the importance of deciphering the underlying physical mechanisms behind the empirical data. In the revised version of the paper, as per the suggestion of the referee, we have tried to emphasize the implications in terms of processes compared to the insight on the major hydrological processes provided in the previous version of the Manuscript (see lines 421-427).

*L127. What are the criteria for the choice of the field deployment? Please, specify.*

Thank you for the comment. Sensors were deployed along the most dynamical part of the tributaries that were observed during the field surveys, in a catchment with a size that makes the field surveys feasible with the human resources available (say, catchment area $< 3\ km^2$). Technical difficulties and the time needed to reach some locations (due to the harsh environment) were thus taken into account in the selection process. The specific location of the sensors was also chosen considering the heterogeneous substrates of the catchment, making it possible to analyze the sensors' behaviour on different type of soils, and ensuring an heterogeneous distribution of the nodes' persistency. These points have been included in the revised version of the manuscript to provide a more in-depth explanation of the criteria for the choice of the field deployment (see lines 149-154).

*L145. This sounds a very short period to me. I understand that practical issues might be arisen but typically we need a longer time period to observe hydrologically processes that often highly variable in time. Si the average rainfall and stream flow in this period comparable to long-term rainfall and stream flow (or compared to the other years where observations are available, since this is a relatively new experimental catchment). I think it's important for the Authors to discuss this issue, explain why (if) they reckon this is a suitable spell and why, and why (if) this is a representative period for derive information on the hydrological functioning of this catchment. Moreover, the should discuss how this short period of time can potentially impact on the results.*

Thanks for the comment. The length of the study period is affected by the characteristics of the study catchment, which is a high relief catchment located in the southern side of the Alps. The basin is snow-covered usually from late November until the end of spring (June). This constraints the time window that can be used to take field measurements and analyze the underlying network dynamics, as winter renews the underlying network dynamics every year. Moreover, the sensors' deployment is highly time-consuming and the set up of all the sensors require at least some months. During the 2019 the underlying climatic conditions were particularly unlucky, as the winter season came earlier than expected (early November instead of early December). At any rate, it is important to stress that though relatively short, the study period covers a wide range of climatic conditions and network configurations, as discussed below. A detailed statistical analysis was performed to analyse the representativeness of the climatic conditions observed during the period of record. Figure 1 of this rebuttal compares the distribution of the daily rainfall depths (h) observed during the reference time window (Sep and Oct of 2019) and that observed in the long term (2010-2020) during the whole period within which the network is dynamical (from July 1 to November 30). The comparison shows that the frequency distribution of the rain depth for our study period matches quite well the corresponding long-term distribution, an instance which suggests that the rain regime during the analyzed period is in line with that driving the observed longer term network dynamics. The significant variations of the underlying hydro climatic conditions during the reference time window is also manifested in the statistic of the cumulated rainfall over 5 and 35 days, which have been shown to be the main drivers of the total active length in the Valfredda (Durighetto et al. 2020). Our analysis indicates that the 5-days cumulative rainfall depth ($h_5$) during the study period ranges from 0 up to the upper quartile of the long-term distribution of $h_5$ within the years 2010 – 2020, indicating that the event variations in rain amounts are well reproduced by the study period. Likewise, for $h_{35}$ the min/max observed range during the period of record (Sep - Oct 2019) is comparable to $2\,\sigma_{h_{35}}$ ($\sigma_{h_{35}}$ being the long-term standard deviation of $h_{35}$). Therefore, while the wet summer could not be included because of the time needed to deploy the sensors and the study period spans only two months, we think it is representative of the longer term network dynamics observed in the Valfredda. A more detailed description of the characteristics of the catchment and of the climatic conditions experienced during 2019 has been added to the revised Ms (see lines 116-127) and the choice of using such a short time window for our study and its impacts on the results has been explained as suggested by the referee (see lines 487-492).

*L184. Where was the sensor placed? In the grass in a convergent zone, I guess, where water was not flowing? Please, give more information on the aspect of sensors deployment in the field.*

Thanks for the comment. The sensor was placed on the grass where water flow was not observed permanently, as the channel activated only during and after precipitation events. In the revised text the deployment in the field of the sensors is described more clearly.

[Figure]

(a) September - October 2019        (b) July - November from 2010 to 2020

Figure 1: Frequency analysis of daily rainfall depth for the study period in 2019 (a) and for the years 2010 to 2020 (b).

**Minor comments**

*L5. I suggest considering the term "customized" instead than "personalized".*
Agreed, thanks. We modified the text as suggested.

*L6. I suggest removing "analysed,".*
Agreed, thanks.

*L6. The expression "nodes' persistency" is not clear without reading the manuscript. Please, clarify.*
Thank you for the comment. We clarified this point by saying "nodes' persistency (i.e. a proxy for the probability to observe water flowing over a given node)".

*L37. Perhaps here the citation to Godsay and Kirchner (2014) fits well. In any case, a more recent references would be a nice complement.*
Thank you for the suggestion. We added more recent references including the following: Floriancic et al., 2018 (DOI: 10.1002/hyp.13302); Godsey and Kirchner 2014 (DOI: 10.1002/hyp.10310); Jaeger et al., 2007 (DOI: 10.1007/s00267-005-0311-2); Lovill et al., 2018 (DOI: 10.1029/2017WR021903).

*L129. "heterogeneous persistencies so as to avoid redundancy in the data." This is not clear to me, please specify.*
Thank you for the comment. What we meant here is that the sensors were deployed close to nodes that had a different degree of persistency in order to assess the consistency of the sensors in various field conditions (from nodes almost always dry to nodes that dries down only sporadically). We specified this point in the revised text.

*L160. Which observations? Please specify. What does "some modelling" mean? Please, explain.*
Thank you for the comment. The models referred to here are described in the paper cited in line 187, Durighetto and Botter 2021 (DOI: 10.1002/hyp.14053). More information was added in the revised text and a brief summary about the model formulation and the main assumptions can be found in Appendix C.

*L176. Building a reliable flow rating curve, especially in mountain catchments, is a challenge. How many points were collected to build the FRC, and what was the range of stream flow values and the resulting goodness of fit measures? In other terms, is the FRC reliable to infer robust stream flow values? Please, explain.*

Thank you for pointing that out, we agree on the comment. The rating curve was built using 7 points with discharge ranging between 9 and 300 l/s and a coefficient of determination $R^2 = 0.99$. These information are included in the revised version of the text.

*L178-179. I understand that the Authors want to give light to the ERC project but this is not the right place. Please, remove.*
Agreed, thanks.

*L199. Did the Authors create a definition of "reliable" for their purposes? The distinction between reliable and not reliable data can be vague. Please, specify what you mean by "reliable".*
Thank you for pointing that out. Non reliable data are meant to be missing data and zeros induced by malfunctioning of the sensors. We acknowledge that the term "reliable" could be considered as misleading and subjective and we removed it from the text. In the revised version we specified the reasons for which some data were not included in the analysis, as indicated above.

*L356. I suggest considering replacing "got" with "became".*
Agreed, thanks.

**REVIEWER N.2**

*This paper reports on what appears to be a pilot project using electrical conductivity sensors to detect presence/absence of flow in a small Alpine basin. I appreciate the difficulty in making these kinds of field measurements, and the paper does a good job of highlighting some of the challenges involved.*
We thank the reviewer for acknowledging the value of this work.

*However, the abstract, discussion, and conclusions make some rather confident claims, without acknowledging the severe limitations of the data and the unusual characteristics of the Valfredda study site. These issues substantially undermine the strong claims presented here.*
We agree that the data have some issues, which are explicitly acknowledged and discussed in the text (Section 2.2.3 and Appendix B). Also, the revised version of the paper emphasized the specific geological features of the study catchment, the duration of the study period, and the impact of these factors on the main results of the paper (see lines 116-127). It is worth noting that the study catchment is not particularly unusual in the end, as most of the headwater catchments in the Dolomitic region - and in general in the southeastern side of the Alps - share similar features in terms of substrate heterogeneity and possible presence of internal karst regions. This was also noted in the revised text (see lines 101-115).

*1 – I am sure that these field data were hard-won, but they span only two months (or maybe only one month – Figures 6 and B2 both refer to "the study period" but one is only about half as long as the other...?), and include only a small handful of precipitation events.*
Thanks for noting that. The study period actually spans two months, as seen in the plots of Figure 5 and explicitly stated in the method section. Figure B2 shows only the period during which data of the sensor $S_{39}$ were corrected. The caption of the figure has been changed accordingly, in order to avoid such misunderstandings.

*It is hard to draw robust conclusions from such limited evidence. The study by Jensen et al. (2019, cited in the references) provides an illustrative contrast, with a much more extensive set of observations, and thus more robust inferences, drawn from a similar number of sensors along a similarly sized channel network (but a longer study period with more precipitation events). I will leave it to the editors to decide whether HESS wants to publish such a limited data set – speaking for myself I would have waited for a more comprehensive picture to emerge.*
We understand that the limited study period may seem problematic. Here, and in the revised manuscript, we explain the reasons why it is so limited, and how this may -or may not- affect our results. The length of the study period is affected by the characteristics of the study catchment which is a small high relief catchment located in the Alps. The basin is snow-covered from late November until the early summer (June). This

constraints the time window that can be used to take field measurements and study the network dynamics (potentially to 4-5 months) and renews the underlying network dynamics every winter. Moreover the sensors' deployment is highly time-consuming and the setup of all the sensors require at least some months. The 2019 fall season was particularly short, as the snow season came earlier (early November instead of early December). At any rate, it is important to stress that, though relatively short, the study period covers a wide range of climatic conditions and network configurations, as discussed below. A detailed statistical analysis was performed to analyse the representativeness of the climatic conditions observed during the period of record. Figure 1 of this rebuttal compares the distribution of the daily rainfall depths (h) observed during the reference time window (Sep and Oct of 2019) and that observed in the long term (2010-2020) during the whole period within which the network is dynamical (from July 1 to November, 30). The comparison shows that the frequency distribution of the rain depth for our study period matches quite well the corresponding long-term distribution, an instance which suggests that the rain regime during the analyzed period is in line with that driving the observed longer term network dynamics. Thus, we do not see any particular reason for which the results of the paper could be strongly impaired by the specific duration of the reference time window. This is especially true if one considers the diversity of the hydrological conditions observed during the period of record. More specifically, the main issues associated to the use of water presence sensors highlighted in the text would not be changed by the use of more data, and the nature of the relationship between the mean persistency of the nodes and statistical properties of the ER signal is unlikely to be significantly modified as well. Likewise, the procedure identified for reconstructing the high frequency space-time dynamics of the stream network using ER data would be the same, even if more data were considered. Also the hysteresis in the $Q$ vs. $L$ relationship would remain, as they are observed within individual events, and across all the events. Of course, it would have been great to have a longer study period. Based on the duration of the snow season and the time needed for the sensors deployment, we argue that the maximum possible duration of the period of record in an Alpine field site of this type could hardly exceed 3 months, after which the winter would freeze the catchment and renew the underlying network dynamics. However, repeating this experiment to get 3 months of data instead of 2 is practically unfeasible, and would imply a huge experimental effort (e.g. another deployment of all the sensors), eventually leading to results that are unlikely to be significantly different from those obtained in this study. All these arguments were clarified in the revised version of the paper (see lines 116-127 and 486-492). In any case we do not see significant discrepancies between our results and the results shown by Jensen et al: in both cases there is a good consistency between ER data and the visual observations of the network, expansions/contractions of the network occur by growth of disconnections within the streams and different types of hysteresis between $L$ and $Q$ are observed.

*2 – The resulting uncertainties are very large (see figure 9), but this is not adequately accounted for in the presentation. The text (line 327) says that b varies by about 1% as the temporal resolution changes, but given that the uncertainty in b can be over 10%, it is actually unknown how stable b really is (or isn't). The text (line 331) even argues for a systematic increase in $R^2$ from 0.485 to 0.522, even though the uncertainty in $R^2$ can be over 20%, making this "systematic" increase statistically meaningless.*

We think the referee is highlighting the fact that the estimate of the value of $b$ and the agreement of the power law model with the data can be dependent on the specific dates of the surveys. This has been further emphasized in the revised text. As per the b exponent, we believe that the pattern shown by the mean scaling exponent $b$ is quite interesting, and that the observed variations of $b$ could be considered as moderate in the light of the scattering of the points in the $L$ vs. $Q$ plan and the simplicity of the power law model. We acknowledge, however, that the pattern of $R^2$ is less meaningful than that of $\langle b \rangle$ owing to the large standard deviation, as suggested by the referee. In the revised version of the paper, we discussed more explicitly this point about the variability of $b$ and $R^2$ across the different sub samples of the data (see lines 376-383; 442-458). Thanks for noting the issue.

*Even these very large uncertainties may be underestimates, because the underlying data are serially correlated, meaning that (for example) few of the points in Figure 8 are statistically independent of one another. From the methods it is unclear whether this has been taken into account, as it should be.*

Thanks for bringing this issue to our attention. Serial correlation is unavoidable in all high frequency joint data set for $L$ and $Q$. Thus we tested the effect of serial correlation of the data in the calculation

of the uncertainty/variance on $b$ and $R^2$ in our case study. To do that, we used a bivariate autoregressive model, fitted to our dataset, to generate a huge synthetic dataset of Q and L data ($10^6$ days) with given autocorrelation functions, which was then used to apply the resampling techniques used in the paper. The results showed that by introducing serial correlation in the generated data, the variance of $b$ and $R^2$ tend to slightly increase only for the smallest resampling frequencies. This indicates that the standard deviations shown in Figure 9 for the smallest resampling frequencies (where effects of the correlation can be observed) should not be underestimated. For the biggest resampling frequencies (T = 4 and 7 days), instead, the (resampled) data does not show serial correlation so the results reported in the paper should not be affected by autocorrelation effects. We thank again the referee for giving us the opportunity to test the effect of the serial correlation in our data. In our case, the integral scale of correlation is about 3.8 days, which is about 1/17 of the study period. This indicates that the dataset provides enough independent data to be representative of the long term network dynamics in the Valfredda catchment (see responses on the previous comments). In light of the issue raised by the referee, we should also clarify the spirit of our analysis. Here we are reproducing a fictitious sampling campaign with sporadic measurements of active length and discharge, assuming that a standard power-law is applied to the sampled data, as typically done in all previous studies where the relationship between $Q$ and $L$ was studied. The standard deviation shown in Figure 9 is a measure of the diversity of model parameters as a function of the sampling dates, not a standard model uncertainty associated to a given data set – which relies on some likelihood function, with possibly correlated residuals. This standard deviation simply resembles the heterogeneity of the fitting of a power-law model done over a set of different sub-samples of the same set of data. In the revised paper we discussed more explicitly how this analysis should be interpreted (see lines 264-268).

*The last main conclusion of the paper is that (lines 411ff): "The mean value of the exponent of the power law relationship between catchment discharge and total active length was found to be almost independent on the frequency of the observational data, which instead had a larger impact on the goodness of fit of the power-law model. When the frequency of the data is lower, the observed values of R2 are, on average, larger..." In view of the vast uncertainties in Fig. 9, these conclusions are reckless. Within the uncertainties, either of these trends could be strongly increasing, strongly decreasing, or zero. There is simply no robust conclusion that can be drawn from the data.*

Thanks for the comment. In the light of the fact that the $R^2$ are not particularly high and the available samples of $Q$ and $L$ are relatively small, we believe that the observed range of variability of $b$ could be deemed as moderate. Therefore, we maintain the point of the relative stability of $b$ for different values of $T$. On the other hand, we recognize that the observed pattern in $R^2$ might not be particularly meaningful, in view of the standard deviation of the estimate shown in Figure 9. This has been emphasized more clearly in the revised version of the paper (see lines 378-383;446-458).

*3 – The limitations of the study site are severe, particularly for analyses of network dynamics. The basic problem is that roughly 80% of the basin seems to have no surficial drainage network at all, consisting instead of talus slopes and moraines. The critical issue here – which is not acknowledged anywhere in the paper – is that this 80% of the basin is still generating discharge (at least some of which is presumably measured at the outlet), but the accompanying network dynamics are invisible because they are occurring beneath piles of rock debris. Outside of the mapped network there appears to be roughly two square kilometers of drainage area with no surficial drainage at all.*

The comment is certainly relevant, and we are grateful to the referee for the insightful input. The referee's guess was an overestimation, as it's very hard to do this kind of estimations just by looking at a map. In the upper Valfredda catchment, 1.7 km$^2$ out of 2.6 km$^2$ (i.e. 65% of the total catchment area) directly drains through the hydrographic network, according to the analyses of the drainage flow paths performed with a high resolution ($20\,cm$) DTM, which are in agreement with our experience in the field. Thus, 35% of the catchment has no surface drainage network. This has been explained in the revised version of the manuscript (see lines 101-109).

*At best, that means that any observations here cannot be compared with the rest of the network dynamics literature, in which the discharge from the whole basin is compared with the flowing stream network across the entire basin. Thus, for example, there is no way to compare Figure 8 with similar diagrams from other*

*studies, because in this case most of the discharge appears to be generated by subsurface flow that is presumably strongly damped and lagged, suppressing the variability in Q (this may account for the sharp vertical lines in Figure 8, for example).*

Thanks for the comment. The drainage density in the upper Valfredda is comparable with that of other study catchments used to study network dynamics (e.g. South Fork of Potts Creek, Fernow, Turbolo). Nevertheless, we recognize that the presence of a karst area which originates a localized spring that releases a quite constant discharge of about 40 $l/s$ (roughly 30 % of the mean discharge) and feeds a perennial stream needs to be taken into account when analyzing the relationship between active length and discharge. This was explicitly done in the revised text (see lines 252-262). The analysis highlighted the limited impact of a permanent water source ($Q_p$) feeding a perennial stream of length $L_p$ on the scattering of the points of the $L - Q$ plot (Figure 8). Also the exponent $b$ of the power law model ended up being similar in the two cases (i.e. with and without the permanent water source).

*The manuscript doesn't confront (or even disclose) this problem anywhere, which is surprising given the abstract's mention of "the diversity of the hydrological behaviour of the study catchment" – by which the paper seems to mean only the two small drainage networks that were studied, not the other roughly three-fourths of the catchment.*

Thanks for the comment (see previous response on the same point). A detailed description of the impact of the portion of the catchment without drainage network on the main results of the paper was added (see lines 364-366; 372-375; 434-435).

*It is virtually a truism in catchment studies that each site has its own idiosyncrasies, but here this particular "uniqueness of place" makes a network dynamics study particularly difficult. Why study network dynamics in a catchment where the great majority of the drainage area has no network at all? Such a site makes it particularly difficult to draw any mechanistic inferences from the observed network behavior.*

The Valfredda catchment is one of the study sites of the ERC project Dynet, because it is quite representative of the headwater catchments in the Dolomitic region. It was used in this study because of the heterogeneity of the substrate, and the availability of long term data about rainfall, discharge, water quality and persistency of the nodes of the network, which lie at the basis of several results presented in the paper (Figures 3 and 4). We respectfully disagree on the claim that mechanistic inferences can hardly be made here, as indicated in the existing literature as the study by Durighetto et al. (2020); in the cited paper three different models were developed and statistically validated in order to describe the dynamics of the active drainage network length (ADNL) starting from the wet length values measured during nine surveys in the field, the atmospheric forcing and the geologic characteristics of the catchment. Both empirical data and the models results showed the influence of the antecedent precipitation on the measured/modelled dynamics of the stream network as much as the geologic features of the study catchment. All these arguments about the choice of the site have been further stressed in the revised text (see lines 151-154; 187-194).

*The manuscript says (lines 367ff): "Network length was found to be more sensitive than discharge to small precipitation inputs: while most rain events induced visible changes in the active channel length, the catchment stream flow was sensitive only to the rain events lasting for several consecutive days (6-9/09, 13-18/10, 20-24/10) and to intense storms (more than 20-30 mm in 9-12 hours)." This is exactly the behavior that one would expect from a field site like this one, with most of the discharge being generated by relatively slow subsurface flow paths over $\sim 80$ percent of the catchment, but with network lengths being measured on the very few surface drainages in the remaining small fraction of the catchment.*

Thanks for the comment, which gives us the opportunity to clarify the issue. In the upper Valfredda catchment analyzed in this paper the karst area does not contribute to the large majority of the discharge, but only to approximately 30% of it (see lines 103-109). In fact, the water that infiltrates in the region where terrain depressions and karst substrates dominate mostly exfiltrate through a permanent spring in the northern portion of the network, originating a constant discharge contribution of less than 40 $l/s$ (while the average observed discharge exceeds 130 $l/s$). The large majority of the total discharge is instead released through the dynamical drainage network mapped in this study. Moreover, we contend that the presence of a permanent water source ($Q_p$) which feeds a perennial stream of length $L_p$ is able to produce the hysteresis in the $L$ vs. $Q$ relationship shown by our data. Within a single intense rain event, the hysteresis depends

on the fact that the active length increases faster than $Q$ in the early stages of the event, while decreases much slower than $Q$ in the recession. Across different events, instead, the hysteresis is generated by shifts in the response of $Q$ and $L$ to different types of rain events. In particular, when rain events are moderate the channel network activates but the amount of water conveyed to the outlet is limited, owing to disconnections and limited flow velocities; conversely, when rain events are more intense the same active length contributes a much larger discharge to the outlet. Anyways, we think that the point made by the referee about the role of a permanent water source for the analysis of $L$ vs. $Q$ relationships is quite interesting. To further investigate the impact of a perennial channel with a constant wet length ($L_p$) supplied by a permanent constant discharge $Q_p$ on the underlying $L$ vs. $Q$ relationship, we added an additional analysis of the data in the revised version of the paper (see lines 253-262; 364-365; 376-378). We considered the total active length as the sum of a constant active length $L_p$ and a dynamical stream network length $L_d$, while the total discharge was assumed to be the sum of a dynamical discharge $Q_d$ and a constant discharge $Q_p$. The dynamical length was estimated to be linked to the dynamical discharge through an exact power law relationship of the type $L_d(t) = a\,Q_d(t)^b$. The observed value of the exponent of the power law model properly fitting the overall values of $Q$ and $L$ was then compared to the $b$ value of the dynamical power law relationship linking $L_d$ and $Q_d$. Figure 8 of the new manuscript highlights the limited difference between the scattering of the points in the two plots, suggesting that the observed pattern is mainly driven by the dynamical components of $L$ and $Q$. Moreover, in both cases the best-fit value of $b$ is similar. The resampling of the data was applied as well to the two paired timeseries $Q - L$ and $Q_d - L_d$ leading to very similar results (Figure 9). These arguments suggest that the presence of the karst area in the northern part of the catchment doesn't affect significantly the fitting of the power-law model and the extent of the observed hysteresis in the $L$ vs. $Q$ relationship. We again thank the referee for raising this important point.

*The manuscript continues (lines 375ff): "In our case study, the standard deviation of the wet length as derived from the sensors' data is 360 m, while the standard deviation of L predicted by the power-law model based on the observed variability of the discharges is only 224 m (about 40% lower). This underestimation is induced by the poor ability of the power law model to capture the observed network dynamics produced by small precipitation inputs." It would rather seem that the problem is that \*no\* model could possibly capture the relationship between the network dynamics in a small part of the catchment, and the discharge generated by completely different mechanisms in the great majority of the catchment.*

As pointed out above, the discharge generated by the unchanneled part of the catchment is about 40 $l/s$, while the mean discharge is more than three times larger (130 $l/s$). The reasons behind the observed hysteresis in the $L$ vs $Q$ relationship are described above, and pertain to differences in the velocity and connectivity of the dynamical streams across different events – or asymmetries in the rate of change of $L$ and $Q$ within individual events. At any rate, we included an in-depth analysis of how the presence of a permanent spring which feeds a perennial stream impacts the discharge vs. active length relationship, as indicated above (see lines 253-262; 372-375; 434-437).

*All of the conclusions concerning the relationship between stream length and discharge (essentially everything after line 10 in the abstract and after line 406 in the conclusions) are based on very thin data from a catchment in which discharge mostly comes from subsurface flow through rock debris (with the result that changes network length in the small fraction of the catchment with surface drainage are unsurprisingly not clearly related to the discharge, which mostly comes from the rest of the catchment).*

As pointed out above, the permanent spring through rock debris contributes to 40 $l/s$ while the average discharge during the study period is 130 $l/s$. See also our previous responses on the same point.

*Thus all of those conclusions are based on very thin data that does not allow straightforward interpretation even in this study catchment, and cannot be extrapolated to the great majority of catchments that lack this particularly exotic geometry.*

Thanks for bringing this issue to our attention. We believe that our conclusions could be of interest for the HESS readers, and not necessarily restricted to the Valfredda site. In fact, hysteresis in the $Q$ vs. $L$ relationship have been previously noted also in other catchments (Zimmer et al. 2017 DOI: 10.1002/2016WR019742; Barefoot et al. 2019 DOI:10.1029/2018WR023877; Jensen et al. 2019) though the underlying reasons were not discussed in-depth. However, the potential impact of the specific features of the study catchment on the results have been emphasized in the revised text, as indicated above (see lines 435-441).

*If those conclusions are excluded – as they really should be, given their weak empirical support and their inherently problematic interpretation (network lengths are not measured in the part of the catchment that generates most of the discharge) – then we have essentially a technical note outlining a new way to deploy conductivity sensors, and conveying some lessons learned from a first deployment of these sensors. That would seem to be a more appropriate way to go, rather than trying to draw strong conclusions about length/discharge relationships from such limited data and such a problematic study site.*

The potential role of the specific features of the study catchment has been better emphasized in the revised text, as per the referee's suggestions. We pondered the possibility of transforming this paper into a technical note, but we don't think it is possible. There is a strong length constraint for HESS technical notes (a few pages) that makes this choice unsuited to this pretty long Ms. In particular, we do believe that the analysis of the joint response of $Q$ and $L$ to the observed precipitation forcing (Figures 7 and 8) should not be removed because they are an important outcome of our field campaign. Moreover, both the reconstruction of the spatial and temporal dynamics of the active stream network (Figure 5) and the analysis of the relationship between the statistical properties of ER signal and persistencies (Figure 4) represent important outcomes of the study, which complement our experimental data set and the discussion of the advantages and disadvantages of the sensors' deployment. Incidentally, the text acknowledges the inherently specific nature of the findings presented in the Ms (e.g. "our results indicate that *in some cases* the use of a bijective $L$ vs. $Q$ function to infer active length changes in catchments where discharge time series are available, *might* lead to significant underestimation of the actual variations of the flowing channel network. *In our case study*, the standard deviation of the wet length as derived from the sensors' data is 360 m, while the standard deviation of $L$ predicted by the power-law model is [...] 35 % lower)."). Nevertheless, in the revised text we emphasized even more that these results might not necessarily apply to all the catchments, and that further investigations are needed to determine the generalizability of the behaviour observed in the Valfredda in the light of the uniqueness of the place (lines 435-441).

*As noted by at least another reviewer, the language would also need work (e.g., "customized" rather than "personalized"), but any revised manuscript is likely to be substantially different so I have not marked those issues in this go-around.*

Thanks for the comment, the text has been polished and the language has been revised. Moreover the new version of the manuscript has been professionally proof read.

---

## Author Response (AR2)

**EDITOR**

*Following up on the assessment of the revised version of your manuscript, there is a need for further modifications and clarifications. A very detailed list of recommendations has been provided by one referee.*

We thank the Editor and the Referee for the long time spent revising our Manuscript and the detailed comments supplied.

*I invite you to consider them very carefully - also the idea of possibly having your contribution wired around the concept of a technical note. The latter would certainly lift some of the main concerns that relate to the extent to which your findings can be considered for drawing more general conclusions in a very specific environment, with a limited amount of data.*

Thank you for the suggestion. We took into consideration the opinion of the Editor and after having exchanged several e-mails with him, we decided to turn our Manuscript into a technical note.

**REVIEWER**

**General comment**

*The second version of this manuscript is an improvement on the first, but severe limitations remain.*

We thank the Referee for acknowledging the work done during the revision of the first version of our Manuscript.

*First of all, the time series is simply far too short to draw the conclusions that are drawn here. The problem is not so much that there are only two months of data, but rather that during those two months there was only one significant hydrological event (meaning: one event that generated a significant change in streamflow), with the result that much of the paper really rests on n=1. Regardless of how many data points were collected at high frequency during that one event, it is still just one event. And there is no basis for assuming that this one event is representative of the behavior of this catchment (note that it is actually relatively small compared to those in the longer-term record in Figure C1 – either in terms of precipitation or the change in network length).*

The limitations implied by the relatively limited duration of the field campaign are acknowledged in the text (lines 383-386). In this new version of the paper, we have tried throughout the Ms. to be clear about the fact that all the conclusions drawn pertain to what emerges from the specific dataset analyzed in the paper (e.g. lines 354-359).

*The authors' response fails to come to grips with this problem; they simply declare that "... the main issues associated to the use of water presence sensors highlighted in the text would not be changed by the use of more data, and the nature of the relationship between the mean persistency of the nodes and statistical properties of the ER signal is unlikely to be significantly modified as well." The first of those statements probably correct, which is why my earlier review indicated that this was more properly a technical note about the process of making these measurements. The second statement, however, is simply a statement of what the authors believe, and there is no evidence showing whether that is true or not.*

The limitations implied by the relatively limited duration of the field campaign are aknowledged in the text (lines 383-386). The paper has been transformed into a technical note as per the suggestion of the Referee.

*The authors do provide a figure comparing the distributions of daily rainfall intensity during their two-month study and during a longer-term record. But this misses the point: how do we know whether the catchment's RESPONSE to the one significant event that occurred can be used to describe the catchment's behavior in general (which depends on far more than instantaneous rainfall intensity)? The problem is not really whether the inputs are representative, but whether the RESPONSE to this one event can plausibly form any basis for generalizations.*

We have removed all the parts of the paper where an attempt was made to relate the specific hydroclimatic conditions experienced during the two months study period and the long-term climatic regime in the Valfredda catchment (lines 122-127 of the previous version our Manuscript).

*Note, for example, that the entire range of discharge that is spanned by these measurements is only about a factor of 10, and in four of the five cases shown in Figure 7 it is less than a factor of two (and sometimes less than 10%!). By comparison, in the study of Jensen et al. 2019 (for example), discharge varied by more than five orders of magnitude.*

There are serious issues with the water balance in the study of Jensen et al. (2019). The ratio between discharge and rainfall is larger than unity in most seasons. High discharges were very likely overestimated in that catchment. The limited variations of $Q$ in the Valfredda, instead, are mostly related to the presence of permanent groundwater sources, which makes the regime persistent (i.e. low $CV_q$).

*In their response, the authors say, "Also, the revised version of the paper emphasized the specific geological features of the study catchment, the duration of the study period, and the impact of these factors on the main results of the paper (see lines 116-127)." This passage, however, consists almost entirely of unsupported claims that these factors have NO impact on the results. It says that the 2-month duration of the study is due to the short snow-free period (which is 6 months, not 2). It says that "the dataset included a wide range of climatic conditions and network configurations", even though Figure C1 in the supplement clearly shows that during this 2-month period (the green shaded region), the network length varied by only about 10-20% of (for example) the range during the following summer. And it contains the very surprising claim that "the study period is reasonably representative of the type of network dynamics experienced by the Valfredda Creek", which again is directly contradicted by Figure C1 in the appendix.*

We have removed the quoted statements from the Ms. Thank you.

*The second major issue pointed out in the previous review is the geological setting. I said in the previous review that "roughly 80% of the basin seems to have no surficial drainage network at all, consisting instead of talus slopes an moraines". That statement is, despite what the authors say, factually correct: the drainage network, such as it is, extends only to a very small fraction of the basin. The authors' response – that 65% of the basin "directly drains through the hydrographic network", based on a DEM analysis – misses the point in two ways.*
*First, the problem here is not that the channel network isn't somewhere below most of the catchment, but rather that it doesn't EXTEND INTO most of the catchment, being confined to a narrow strip along the western edge, consistent with the rest of the network being buried by moraines and talus slopes, which are visually obvious in Figure 1 (if you enlarge it, you can see the individual talus blocks). This network has been mapped as extending to within a few meters of the divide in the northwest direction (see Figure 1b), but stopping more than a kilometer from the divide in the northeast direction. In other words, as stated in the previous review, most of the basin has no surficial drainage network at all.*

We acknowledged the uneven distribution of the stream network in our catchment (lines 94-96). In the revised version we have also pointed out that the analysis presented relies on a high resolution DTM (lines 96-98).

*Second, the DEM analysis assumes that subsurface flow follows the topographic gradient, which is at best unproven at Valfredda and is known to be false in many similar karstic catchments (and particularly karstic catchments buried by talus and moraines!).*

Thank you for the comment. In lines 94-107 we have provided a fairer description of the geologic characteristics of our study catchment removing all the unnecessary assumptions.

*I appreciate that this geological setting may be typical of the Dolomite Alps. But it is still a very poor setting for trying to understand the processes that underlie network dynamics, because most of the network has been buried by talus, and thus most of the stream discharge is generated from areas where the network dynamics are unobservable.*

We acknowledged the complexity of studying the hydrological processes of the Valfredda catchment (lines (lines 102-103) and clarified that the results obtained depend on the specific dataset analyzed and on the peculiar characteristics of the site (lines 354-359). Thank you.

*The authors say that the 35% of the basin that doesn't even slope toward any channels is the source of several perennial springs, but there is no valid basis for this claim (which would require tracer data). There are perennial springs, and there is an internally draining part of the catchment. No evidence has been presented showing that one is the source of the other. (In this regard, the authors' letter is not even consistent with the manuscript; the letter claims that this "karst area" feeds one spring with "a quite constant discharge of about 40 l/s", whereas the manuscript says that this area feeds "a couple of localized springs with a seasonally variable discharge that ranges from about 60 l/s in the last spring to 35 l/s in the fall".)*

We thank the Referee for the comment. We have removed the claim that linked the spring to the karst region of the catchment. Moreover, as most of the highlighted problems concern intrinsic characteristics of our study site and the length of the period analyzed (which can not be changed by definition), we decided (after an e-mail exchange with the Editor) to turn our Manuscript into a technical note. In this way we could focus the attention on our catchment and its specificity without drawing general conclusions.

*The authors say that "the text has been polished and the language has been revised. Moreover the new version of the manuscript has been professionally proof read." The authors may wish to reconsider who they employ in this regard. To be clear, English is a tricky language, but there are many errors here that a professional should catch. Here is a sample (probably not complete) of some of the most obvious problems:*
*"important ecosystems services"*
*"Whol" (her name is spelled "Wohl")*
*"various degree of flow intermittency"*
*"discharge measurement were taken"*
*"a quite widespread phenomena" (should be phenomenon)*
*"the sensors' net were screwed on rock emergencies"*
*"a smaller mean inter-sensors distance"*
*"is greater or equal than"*
*"in the Appendix C"*
*"for each time of the surveyed period"*
*"the dashed line represent the threshold"*
*"these kinds of catchment"*
*"an higher persistency"*
*"an higher average intensity"*
*"it was responsible to increase"*
*"the trend observed during the event occurred"*
*"precious information"*
*"the ER sensors signal provides"*
*"Out data indicated the presence"*

We thank the Referee for the careful research of the English typos and mistakes still present in our Manuscript. We apologize for the inconvenience, we have sent a formal complaint letter to the company who did the copyediting and we were more scrupulous and careful while revising the new version of the document.

**More specific comments**

*L 39: "However, this method also proved to be highly time-consuming for relatively small catchments. Recent technological advances in the field of environmental sensing provide a good opportunity to support the observational reconstruction of stream network dynamics." But Figure C1 rather clearly shows that at least in this case, the direct measurements were more informative than the brief period of high-frequency automated measurements. It is inconsistent to claim that the automated measurements are preferred because they take less time, and then to claim that this 2-month record of automated measurements could not be repeated in another year because it would be too much work.*

In the quoted text we did not mean to say that automated measurements should be preferred to field surveys because they take less time. On the contrary, the use of small sensors deployed on a mountain catchment and subjected to a huge diversity of climatic conditions requires even more attention and effort that those required by direct surveys: the field surveys that were necessary to collect all the data and check the sensors status were challenging and time consuming, not to mention the time spent analyzing and correcting the data. However, at the moment, remote sensors are one of the few alternatives that allow high frequency measurements of the active length to be taken. In fact data obtained from field surveys can not provide wet length values on an hourly or daily timescale. Figure C1 shows the active length estimated by the empirical model described in Durighetto et al. (2020) and the field surveys carried out on the study catchment to

calibrate and validate that model. That plot can not be used to assume whether direct measurements are more or less informative than automated measurements, as a number of independent observations at a low temporal resolution cannot surrogate continuous observations of network dynamics during single rainfall events. At any rate, we decided to turn our Manuscript into a technical note because of the intrinsic characteristics of our study site, which make impossible the use of ER sensors for more than a couple of months per year.

*L 83-89: This is unnecessary. (Readers will already know, for example, that a set of conclusions closes a scientific paper...)*

Agree, we have removed that paragraph.

*L 117: you have a six-month snow-free period, so that's not a valid reason for only having two months of data. As stated in the previous review, I understand that getting good field data in such alpine settings is difficult. But that's why good data often require multiple field seasons.*

See our previous answer on the same point, thank you.

*L 125: as stated above, there is no evidence showing that the network dynamics, per se, during this period were representative of the catchment's behavior over any other time interval.*

We have removed that statement, thank you.

*L 341: "the dynamics of L were mainly driven by precipitation, the temporal pattern of which differs from the corresponding streamflow dynamics because of the non linearity of rainfall-runoff mechanisms". This is a non-sequitur. Precipitation and streamflow dynamics will be different even in linear rainfall-runoff mechanisms!*

We modified the quoted sentence as follows "the dynamics of L were mainly driven by precipitation, the temporal pattern of which always differs from the corresponding streamflow dynamics".

*L 369: this p-value would only make sense if the data points were all independent and the residuals were normally distributed. Neither is likely to be true here.*

Thank you for pointing that out, we have removed the p-value.

*L 380: what is called "range of variability" here is not range, but standard deviation. And the increase in variability is exactly what one would expect from the loss of degrees of freedom with the sparser sampling.*

Thank you for the comment. We have changed the quoted paragraph and moved it to an Appendix.

*L 434: "It is worth noting that the presence of the localized springs did not affect the performance of the power-low model, indicating that the presence of karst areas might not be the cause of the observed hysteresis in the L vs. Q relation." I needs to be recognized that no evidence is presented to show that the water that infiltrates in the "karst areas" (whatever exactly these are... as far as I can tell the entire basin is karstic) emerges from the "localized springs" (and not elsewhere, or not \*also\* elsewhere). Likewise no evidence is presented to show that the flow from the localized springs originates from (let alone originates \*entirely\* from) the "karst areas".*

Thank you for the comment. We removed the quoted sentence.

*L 454: "Nevertheless, when the mean interarrival between the observations increases, the variability across the samples is more pronounced and the chances that the experimental (L;Q) pairs don't exhibit a clear power-law trend also increases, as shown by the growth of the standard deviation of R2 with T. This suggests that the goodness of fit of the power-law model can be strongly dependent on the specific timing*

*of the field surveys in which active length and discharge are evaluated, making the observed pattern of R2 poorly informative." This is correct, but it is also Basic Statistics 101, and thus should not be presented as if something notable has been discovered here. Or if there is something happening here that is not obvious from a basic understanding of statistics, that distinction should be pointed out.*

We thank the Referee for the comment. We have modified the quoted sentence as follows: "This shows, as statistics suggests, that the goodness of fit of the power-law model can be strongly dependent on the specific timing of the field surveys in which active length and discharge are evaluated, making the observed pattern of $R^2$ poorly informative." Moreover we moved the description of the resampling and of the results obtained from the main text to the Appendix.

*L 463: "The mean intensity of the ER signal, and the exceedance of a suitably selected intensity threshold were found to be highly correlated with the persistency of the network nodes. This suggests that ER sensors signal provides statistically meaningful information on the hydrologic status of different nodes of the river network." From statements like these (also in the abstract), readers would reasonably infer that the "persistency of the network nodes" is something that was actually measured. That is not the case. It is a modeled quantity, not real data per se. There are no measurements of network persistency (at least, none are presented here).*

We thank the Referee for the comment. In the Manuscript there is an entire subsection dedicated to this topic (Section 2.3.1) where we explain our choice to use modeled persistency values instead of directly measured values. During the study period we frequently performed surveys in the field, however we did not map the entire stream network at every survey because collecting the data and checking on the sensors status on this alpine catchment proved to be very time-consuming. The experimental data alone could not be used to calculate the persistency of the nodes so we decided to estimate it on a modeling basis.

*L 478: "an in-depth analysis..." Sorry, but one cannot develop an "in-depth analysis" from just one substantial hydrological event!*

Thank you, we have removed the word "in-depth".

*L 490: "we believe that more extensive field campaigns would not significantly modify the main conclusions of this study..." This may be accurate as a statement of the authors' beliefs, but it is not supported by evidence.*

We have removed the quoted statement from the Ms.

*L 492: "... study, as the features outlined in the paper emerged systematically from the data collected during a sequence of events, which were few in number but characterized by heterogeneous hydroclimatic features." Figure C1 shows that, in contrast to this statement, the brief period studied here was characterized by relatively \*homogeneous\* behavior, compared to the other snow-free periods in that longer record.*

We have removed the quoted statement from the Ms.

---

## Author Response (AR3)

**EDITOR**

*Following up on a review of the latest version of your manuscript, I consider your manuscript now receivable for publication in HESS. Prior to the final processing steps, please have a look at the figure numbering, which obviously jumps from 6 to 8 as a result of a reorganisation of the manuscript. This would have to be corrected before final acceptance of your contribution.*

We thank the Editor and the Referee for their decision. The figure numbering has been revised.

**REVIEWER**

*The current version of the manuscript has been substantially reconfigured as a technical note. The authors have also qualified some of their conclusions. While they have not gone as far as I think they should have, the manuscript is now in a state where I think the editor should just decide whether it is now suitable for the journal. (One thing that obviously must be fixed is the figure numbering, which jumps from 6 to 8.)*

Thank you. The figure numbering has been fixed as suggested.